# Jawsamycin exhibits in vivo antifungal properties by inhibiting Spt14/Gpi3-mediated biosynthesis of glycosylphosphatidylinositol

Yue Fu [1,5], David Estoppey [2,5], Silvio Roggo[2,5], Dominik Pistorius[2,5], Florian Fuchs [2,5], Christian Studer[2,5], Ashraf S. Ibrahim[3,4], Thomas Aust[2], Frederic Grandjean [2], Manuel Mihalic[2], Klaus Memmert[2], Vivian Prindle [1], Etienne Richard[2], Ralph Riedl[2], Sven Schuierer [2], Eric Weber[2], Jürg Hunziker[2], Frank Petersen[2], Jianshi Tao [1✉] & Dominic Hoepfner [2✉]

Biosynthesis of glycosylphosphatidylinositol (GPI) is required for anchoring proteins to the plasma membrane, and is essential for the integrity of the fungal cell wall. Here, we use a reporter gene-based screen in *Saccharomyces cerevisiae* for the discovery of antifungal inhibitors of GPI-anchoring of proteins, and identify the oligocyclopropyl-containing natural product jawsamycin (FR-900848) as a potent hit. The compound targets the catalytic subunit Spt14 (also referred to as Gpi3) of the fungal UDP-glycosyltransferase, the first step in GPI biosynthesis, with good selectivity over the human functional homolog PIG-A. Jawsamycin displays antifungal activity in vitro against several pathogenic fungi including Mucorales, and in vivo in a mouse model of invasive pulmonary mucormycosis due to *Rhyzopus delemar* infection. Our results provide a starting point for the development of Spt14 inhibitors for treatment of invasive fungal infections.

[1] Genomics Institute of the Novartis Research Foundation, 10675 John Jay Hopkins Drive, San Diego, CA 92121, USA. [2] Novartis Institutes for BioMedical Research, Novartis Pharma AG, Forum 1 Novartis Campus, CH-4056 Basel, Switzerland. [3] The Lundquist Institute for Biomedical Innovations at Harbor-University of California at Los Angeles (UCLA) Medical Center, Torrance, CA 90502, USA. [4] David Geffen School of Medicine at UCLA, Los Angeles, CA 90095, USA. [5]These authors contributed equally: Yue Fu, David Estoppey, Silvio Roggo, Dominik Pistorius, Florian Fuchs, Christian Studer. ✉email: jtao@gnf.org; dominic.hoepfner@novartis.com

nvasive human fungal infections are associated with high morbidity and mortality rates[1]. The impact of these diseases is underappreciated but the annual death rate due to serious fungal infections exceeds that caused by tuberculosis or malaria[2]. Despite increasing numbers of affected patients with life-threatening infections and the concomitant increased burden on public healthcare systems, there remains only three main classes of established antifungal agents to treat systemic infections. These are the polyenes, azoles, and echinocandins which act on the fungal plasma membrane, its biosynthesis pathway or cell wall components, respectively[3]. These classes of anti-fungals suffer from restrictions in route of administration, spectrum of activity, poor bioavailability in target tissues, undesirable drug interactions, limited therapeutic index and the emergence of drug resistance[2]. In addition, due to a change in the affected patient populations (immunocompromised patients, premature born, and elderly people), the spectrum of isolated pathogens is shifting to species which are resistant to available anti-fungals. For example, there is no reliable medical therapy for infections caused by species of molds such as Rhizopus[3]. These concerns are exacerbated by the lack of development of novel antifungal agents in the current clinical pipeline against fungal diseases.

Recently chemical biology findings have highlighted the glycosylphosphatidylinositol (GPI) biosynthesis pathway as a promising novel target pathway to treat life-threatening fungal infections. GPI biosynthesis is a conserved process, required for anchoring proteins to the plasma membrane of fungi and thus essential for the integrity of the fungal cell wall in yeasts and molds[4]. In addition, disruption of GPI bionsynthesis is detrimental to fungal cells as GPI anchor maturation impairment disrupts proteostasis in the endoplasmic reticulum causing potentially lethal cellular stress. In addition, lack of GPI-anchored proteins at the cell surface interferes with cell wall integrity eventually leading to lysis while loss of the GPI-protein coat has been shown to expose the immunogenic fungal β-glucans and trigger an inflammatory host response[4–7]. Furthermore, sequence homology between mammalian and fungal genes in the GPI pathway show modest conservation raising the chances of finding fungal-selective molecules with good therapeutic index[8].

Therapeutic exploitation of this pathway, however, is hampered by lack of structural information. To date, no crystal structure of any GPI enzyme has been solved. It is thus not a surprise that inhibitors against only two targets in the entire pathway have been identified. Using phenotypic screens and chemical genetics, a set of synthetic compounds have been identified as inhibitors of Gwt1[5,9], a protein involved in the acylation of inositol early in the GPI pathway, and a natural product chemotype as an inhibitor of Mcd4[10,11], an ethanolamine phosphotransferase later in the process (all reviewed in detail by Mutz and Roemer[12]). Despite this limited chemical armamentarium, the combined experimental data collected using these compounds strongly supports the GPI pathway as an attractive starting point for novel antifungal therapies. Chemical inhibition of Gwt1 has been achieved with broad fungal species selectivity without interfering with the mammalian orthologue and second generation molecules have entered clinical testing[13–15]. The compounds targeting Mcd4 display an even broader spectrum of antifungal activity however it appears that they also inhibit the mammalian homolog[10,11,16]. Thus further attempts to simplify the natural product scaffold and increase fungal selectivity will be required to fully assess the therapeutic potential of Mcd4 inhibition. Nevertheless, chemical interference at both nodes supported the GPI pathway as a promising novel antifungal target.

Here we present a phenotypic screening approach aimed at identifying novel inhibitors of the GPI biosynthesis pathway in fungi. The screen identified jawsamycin as a potent primary hit.

Jawsamycin is an oligocyclopropyl-containing natural product for which broad activity against fungi but not bacteria has been described but the target was unknown[17]. Using a suite of genetic assays, we have shown that jawsamycin selectively and potently targets fungal Spt14 (alias name: Gpi3) but not the human PIG-A homolog. A focused set of jawsamycin analogs were prepared and profiled against four fungal pathogens identifying a tight structure activity relationship. Jawsamycin shows potent and fungicidal activity against pathogens of the Mucorales order and initial in vivo experiments support further development of jawsamycin as a novel antifungal lead compound.

## Results

**Identification of jawsamycin as a GPI pathway inhibitor.** To identify novel inhibitors of the GPI pathway, we adapted a previously published reporter construct expressing the small *Gaussia princeps* luciferase gene fused to the GPI-anchoring signal of *Candida albicans PGA59*[18] (Fig. 1a). Expression of this construct from the galactose-inducible GAL1/10 promoter in the non-pathogenic yeast *Saccharomyces cerevisiae* allowed to monitor cell surface-bound luciferase in relation to non-anchored, secreted luciferase signal in the medium in case of GPI pathway modulation. Testing the assay with two antifungal agents and one known GPI inhbitor[19] revealed an increase of signal in the medium paralleled by a reduction of signal of the cell pellet with the GPI inhibitor only (Fig. 1b, Supplementary Fig. 1). This suggested a lack of anchoring of secreted reporter protein as expected. To benchmark if this reporter was specific for inhibition of the GPI pathway, we tested a panel of compounds with known, diverse mechanism of actions (Fig. 1c). Based on this set of compounds, it was confirmed that the signal of non-anchored luciferase in the medium increased only when GPI anchor biosynthesis was inhibited. Using this 1536-well supernatant signal assay, we screened a focused set of 12472 compounds selected from the Novartis natural product library at a fixed dose of 10 μM (Fig. 1d). The screen had acceptable quality with an overall z′ score of 0.42–0.54. Among the top hits were structural variants of known Gwt1 and Mcd4 inhibitors as well as two isolation batches of a natural product named jawsamycin. Jawsamycin (also known as FR-900848) is a structurally unique oligocyclopropyl-containing natural product and has been proposed to have potent antifungal activity[17,20] (Fig. 1e). To support validation and further testing, we optimized fermentation conditions using *Streptomyces luteoverticillatus* to a yield of 10 mg/l in shake flasks and 5 mg/l in large scale fermenters (see Methods section) and retested the new purified batch in dose-response fashion in the reporter gene assay (Fig. 1f). Concentration-dependent decrease of luminescence in the cell-pellet fraction by jawsamycin was confirmed and a half maximal inhibitory concentration (IC$_{50}$) of ~7 μM measured. This validated jawsamycin as a GPI pathway modulator.

**Chemogenomic profiling points to first node in the GPI pathway.** We next sought to discover the target of jawsamycin. Potent growth-inhibitory activity with an IC$_{50}$ of ~800 nM (Fig. 1g) made it a suitable compound to investigate the mechanism of action by chemogenomic profiling using the *S. cerevisiae* heterozygous and homozygous deletion collections. Haploinsufficiency profiling (HIP) and homozygous profiling (HOP) are gene-dosage dependent methods that assesses the effect of compounds against potential targets encoded by the *S. cerevisiae* genome[21,22]. HIP indicates pathways directly affected by the compound. HOP (both gene copies deleted) indicates synthetic lethality and identifies compensating pathways to those directly affected by the compound.

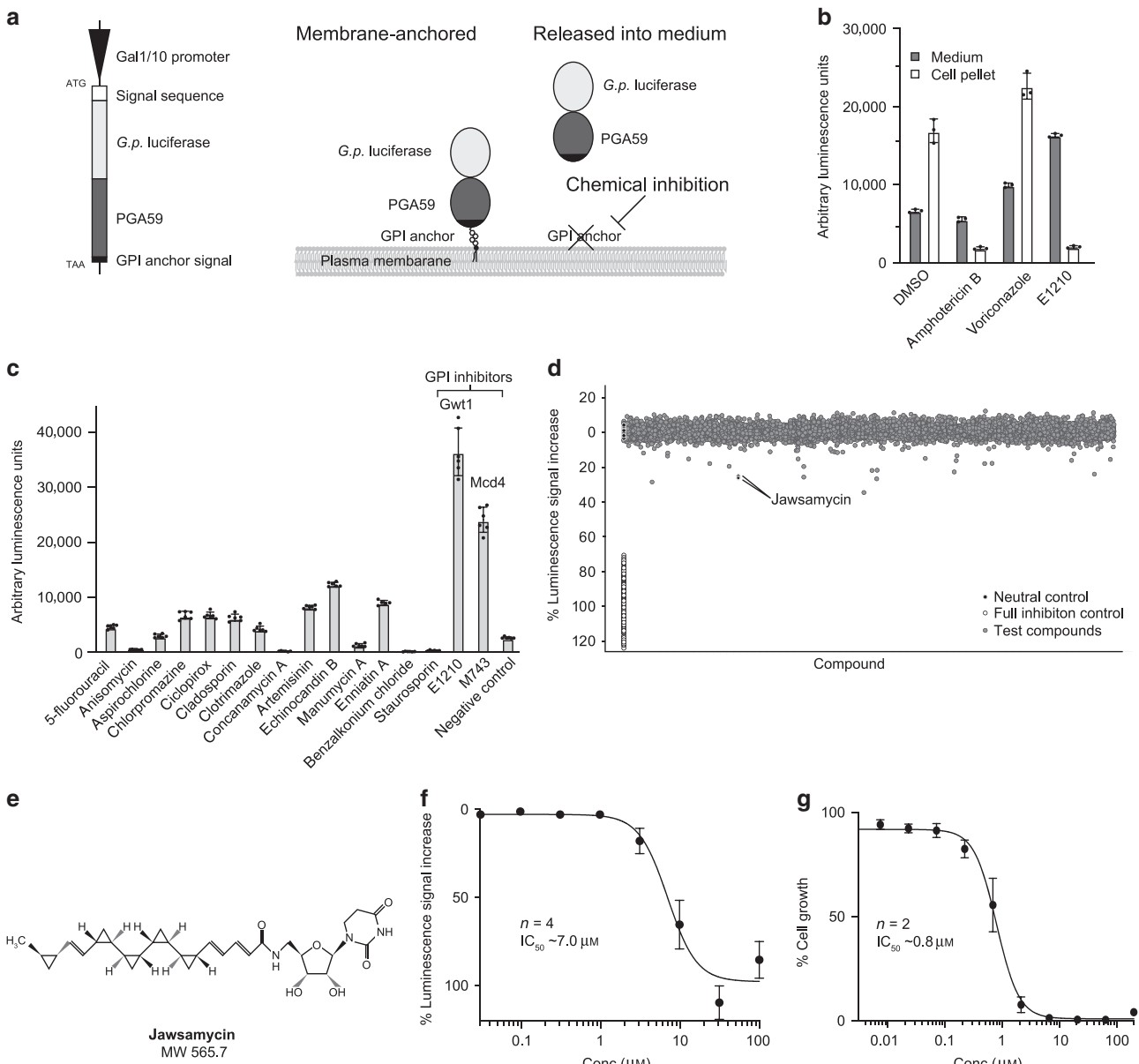

**Fig. 1 Identification of jawsamycin in a screen for GPI biosynthesis inhibitors. a** Schematic representation of reporter gene construct and assay principle. **b** Characterization of signal distribution in medium and pellet upon growth inhibition. Approximate $IC_{50}$ compound concentrations determined in pre-experiments were used in the test (1 μM for voriconazole and E1210, 10 μM amphotericin B). **c** Testing of assay specificity using inhibitors against different known targets and pathways. **d** Primary assay data of 12472 natural products screened at 10 μM. **e** Structure of jawsamycin, a key hit in the screen. **f** Dose-response validation of jawsamycin in the reporter gene assay. **g** Dose-response validation of jawsamycin against growth of *S. cerevisiae*. Bars in panel **b** depict the mean of 3, in panel **c** the mean of 6 measured wells, dots in panel **f** and **g** depict the mean of 2 replicates; in all panels error bars depict +/− one standard deviation. All raw data is available in the Source Data file.

Two independent HIP assays reproducibly identified, *GPI1*, *GPI2*, *GPI15*, and *SPT14*, as hits in essential genes with the best z-scores (Fig. 2a). These four components are 4 of 6 subunits which comprise a complex to transfer UDP-GlcNAc to phosphatidylinositol, the first intermediate in the synthesis of glycosylphosphatidylinositol (GPI) anchors[23]. *PIG19*, encoding one additonal, essential subunit displayed moderate hypersensitivity in the HIP assay, the sixth subunit encoded by *ERI1* was not resolved as this strain is not part of the standard yeast deletion collection. Hypersensitivity of these hits was unique to jawsamycin and has never been observed in chemogenomic profiling experiments of more than 4000 diverse other compounds[22]. Not only does this underline the biological relevance of the hits, but also suggests that the used reporter assay successfully enriches for compounds modulating the GPI pathway. Re-testing by dose-response experiments of the individual strains confirmed the finding from the genome-wide profiling experiment (Supplementary Fig. 2a). GPI-anchored proteins receive the GPI anchor as a conserved posttranslational modification in the lumen of the endoplasmic reticulum (ER). After anchor attachment, the GPI anchor is structurally remodeled to function as a transport signal that actively triggers the delivery of GPI-anchored proteins from the ER to the plasma membrane[8] where many have essential functions in cell-wall synthesis[7]. Inhibition of GPI biosynthesis has been shown to lead to disrupted proteostasis in the ER triggering the activation of the unfolded protein response (UPR)[5]. In agreement with inhibition of GPI biosynthesis by jawsamycin, the two key components for activation of the UPR, *HAC1* and

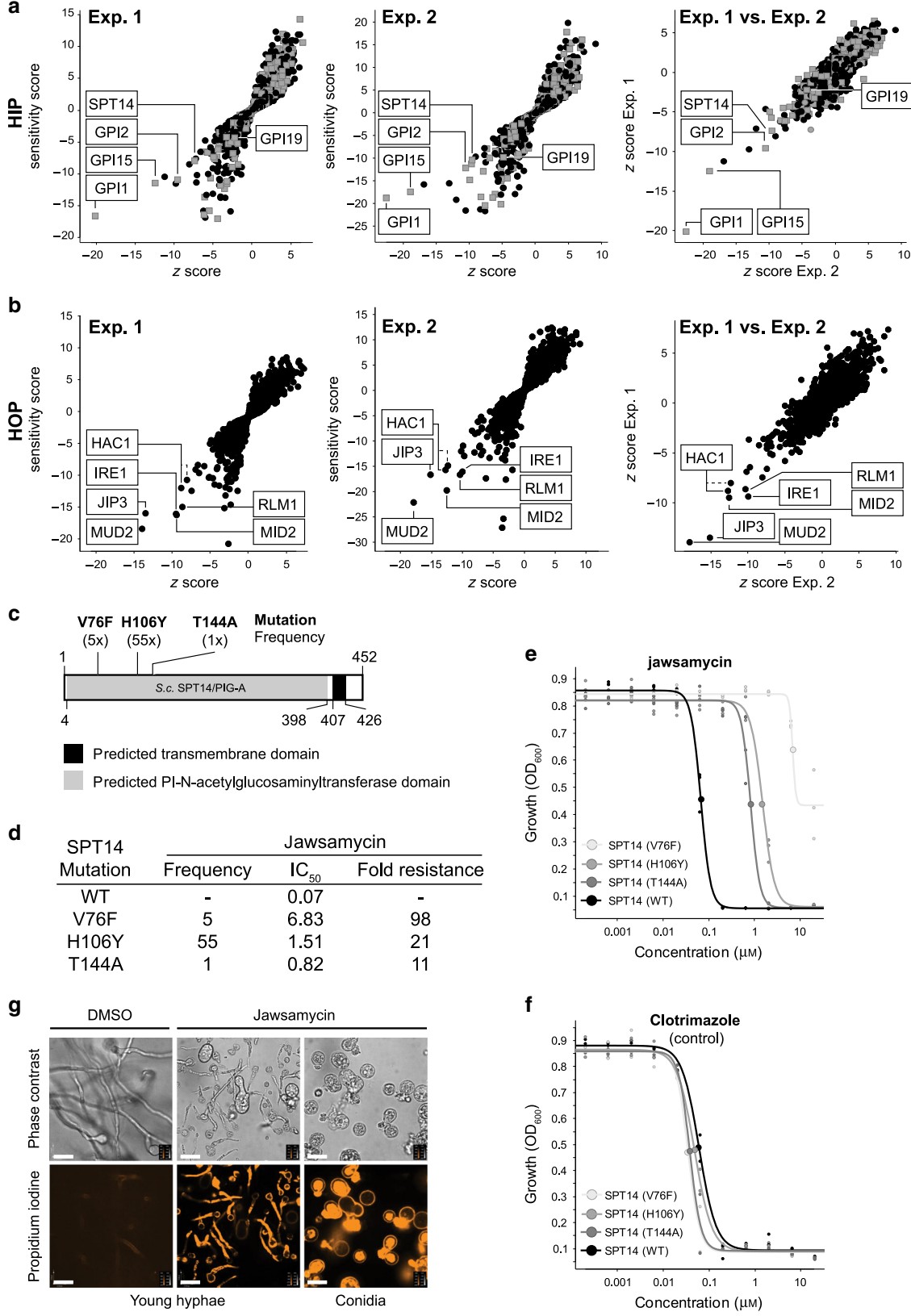

IRE1 scored as prominent hits in the HOP experiment (Fig. 2b). As for the hits from the HIP experiment, hypersensitivity of *hac1* and *ire1* deletion strains was confirmed in single strain experiments (Supplementary Fig. 2b). Therefore, unbiased chemogenomic profiling experiments identified heterozygous

and homozygous deletion strains in agreement with inhibition of GPI anchor biosynthesis.

**Mutations in *SPT14/GPI3* confer jawsamycin resistance**. Chemogenomic profiling allowed us to postulate that jawsamycin

**Fig. 2 Identification of the target of jawsamycin. a** Two independent haploinsufficiency profiling (HIP) experiments identify strains compromised in genes encoding the phosphatidylinositol N-acetylglucosaminyltransferase complex (GPI1, GPI15, GPI2, and SPT14) as selectively and reproducibly hypersensitive against Jawsamycin. GPI19, encoding an additional subunit of this complex scored some hypersensitivity, response of ERI1 also associated with this complex could not be recorded as this strain is not part of the standard yeast deletion collection. **b** Two independent homozygous profiling (HOP) experiments with jawsamycin reproducibly identify synthetic genetic nodes in the unfolded protein response (HAC1, IRE1), cell wall integrity (MID2, JIP3, RLM1) and the pre-mRNA splicing component gene MUD2. The profiles plot sensitivity (y-axis) vs. z-score (x-axis) and the entire dataset is available in the Source Data file. The z-score represents a measure for uniqueness of a hit in relation to >3000 tested, diverse chemical compounds (see materials and methods section). Gray and black dots represent strains with deletions in essential and non-essential genes, respectively. **c** Predicted protein domains and relative position of the identified, resistance-mediating mutations. **d** Frequency, $IC_{50}$ concentrations and fold resistance of identified mutant clones as calculated using the curves depicted in panel e. **e** Dose-response validation of identified mutations introduced into non-mutagenized wild-type cells; $n = 3$, curves depict the calculated mean. **f** Testing of identified mutations against the chemically distinct ergosterol inhibitor clotrimazole indicates that observed resistance to jawsamycin is selective as no obvious $IC_{50}$ shift could be observed. $N = 3$, curves depict the calculated mean. **g** Propidium iodine staining of R. oryzae samples indicates fungicidal phenotype when incubated with jawsamycin. Images are representative examples of two independent experiments where more than 20 fields of interests where visually inspected and that consistently displayed the same effects. Size bar represents 50 µM.

interfered early in the GPI biosynthesis process likely at the level of GlcNAc-phospatidylinositol formation, but did not provide the resolution to identify the primary target. We therefore pursued an induced, unbiased mutagenesis screen aiming at the identification of resistance conferring mutations. In the past, this procedure has successfully identified the primary target and binding pocket of bioactive compounds[24–26]. Plating a total of $2 \times 10^7$ mutagenized haploid and diploid cells onto growth-inhibitory concentrations of jawsamycin, we identified 18 haploid and 53 diploid resistant colonies that were streaked out onto new selective plates to confirm resistance. We subjected 12 resistant diploid clones to full genome-sequencing with an average 60-fold coverage. Selecting for non-synonymous single-nucleotide polymorphisms (SNPs) in coding regions identified SPT14 as the gene with the highest SNP frequency revealing mutations in 10 out of 12 cones. Eight clones had a heterozygous H106Y mutation, one clone a V76F mutation and one clone a T144A mutation (Fig. 2c). Although the H106Y mutation was identified with the same underlying DNA mutation in all 8 clones, differences in genome-wide SNP pattern of the eight mutants suggested independent events. Heterozygous mutations supported the idea that the resulting resistance was dominant. No mutations were identified in GPI1, GPI2, and GPI15 and the findings from the genome-wide sequencing confirmed by focused Sanger sequencing. Next, we expanded SPT14 analysis to all other resistant clones identifying a total of 55 H106Y, 5 V76F, and one T144A mutations. To confirm that the three mutations in SPT14 were linked to the observed resistance against jawsamycin, we introduced these three mutations into fresh wild-type cells and confirmed $IC_{50}$ shifts in the range of 10–100 fold (Fig. 2d, e). Resistance was specific as no $IC_{50}$ shift was recorded against the chemically unrelated ergosterol inhibitor clotrimazole (Fig. 2f). Therefore, unbiased genome-wide mutagenesis identified three point mutations in SPT14, the catalytic subunit of the GlcNAc-phospatidylinositol transferase complex that confer resistance against jawsamycin.

**Potent and cidal action against pathogenic fungi by jawsamycin.** Targets in the GPI biosynthesis pathway have previously been proposed as excellent therapeutic intervention points due to interference with cellular fungal integrity at multiple levels as previously described[5]. Our experiments in yeast have supported potent and rapid growth arrest upon exposure to jawsamycin and HOP profiling identified synthetic lethality with key components in the ER-stress pathway. Encouraged by these findings, we assessed the potential of jawsamycin to affect viability of pathogenic fungi. We subjected 18 different pathogenic fungi to antifungal susceptibility testing according to the Clinical and Laboratory Standards Institute (CLSI) guidelines and recorded the minimal effective concentration (Table 1). Jawsamycin

**Table 1 Fungicidal activity of jawsamycin.**

| Pathogen class | Pathogen name | ATCC ID | MEC (µg/ml) |
|---|---|---|---|
| Yeast | Candida albicans | 24,433 | 1.2 |
| | Cryptococcus neoformans | 36,556 | 2.0 |
| Mold | Aspergillus fumigatus | MYA-3627 | 0.5 |
| | Aspergillus flavus | 16,883 | ≤0.008 |
| | Aspergillus lentulus | MYA-3566 | 0.25 |
| | Aspergillus nidulans | 38,163 | >4.0 |
| | Aspergillus terreus | MYA-3633 | 0.031 |
| | Fusarium oxysporum[a] | MYA-35 | ≤0.008 |
| | Scedosporium apiospermum[a] | MYA-3635 | 0.063 |
| | Scedosporium prolificans[a] | 200,543 | 0.063 |
| Mucorales | Rhyzopus delemar[a] | MYA-4621 | 0.016 |
| | Rhyzopus oryzae[a] | MYA-3791 | ≤0.008 |
| | Absidia corymbifera[a] | MYA-3781 | ≤0.008 |
| | Absidia corymbifera[a] | 66,996 | ≤0.008 |
| | Mucor circinelloides[a] | MYA-4072 | 0.016 |
| | Mucor indicus[a] | MYA-3789 | ≤0.008 |

[a]Fungal species insensitive to voriconazole and echinocandin.

exhibited antifungal activity with broad spectrum and particular potency against Fusarium spp., Scedosporium spp., and Mucorales fungi including Rhizopus oryzae, Absidia corymbifera, and Mucor circinelloides. It is worth noting that these fungal species are generally insensitive to current licensed antifungal agents. Comparing the published SPT14 sequence homologs of representative pathogenic fungi, revealed conserved residues identified to yield resistance when mutated in S. cerevisiae with either identical or similar amino acids (Supplementary Fig. 3). One notable exception was the sequence of C. neoformans that carried the equivalent of the S.c.T144A mutation, likely explaining the high minimal effective concentration (MEC). However, the relatively low potency against C. albicans was less obvious since the overall similarity to S.c. Spt14 was high (57%) and all three residues conserved in the "sensitive state" (Supplementary Fig. 3).

Inhibition of GPI biosynthesis has been demonstrated to not only be cytostatic but fungicidal to fungal cells[14], a desired characteristic of antifungal agents. As jawsamycin has been

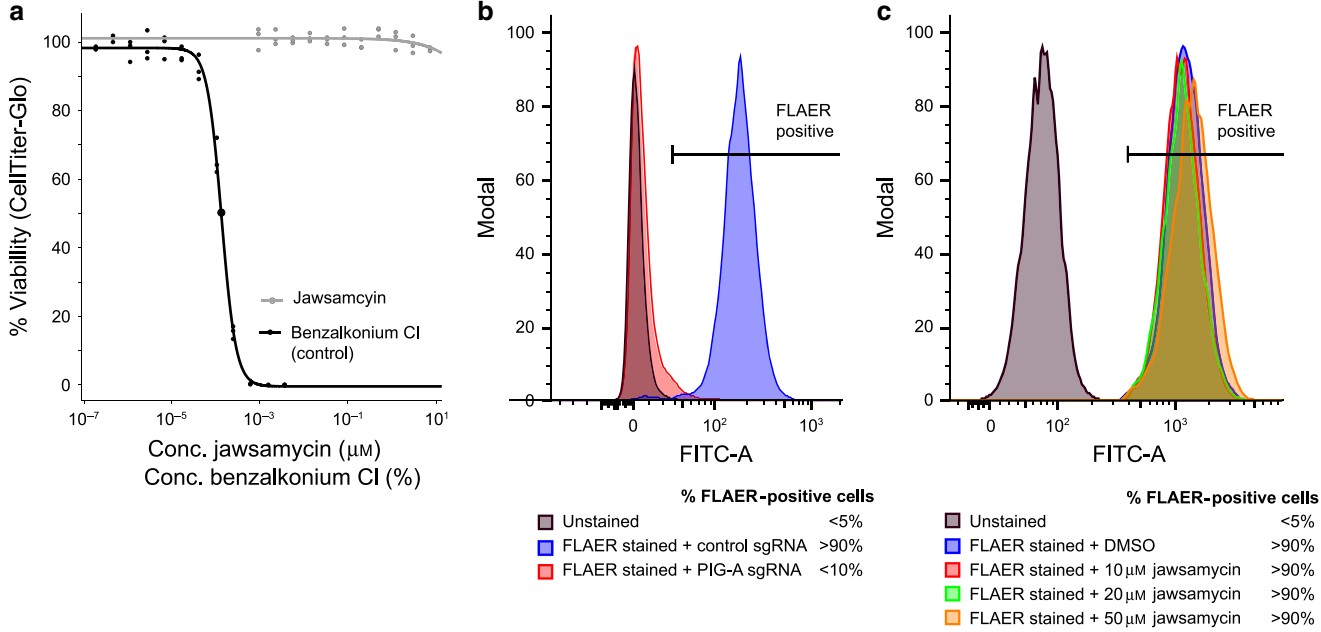

**Fig. 3 Jawsamycin does not display activity against the human *SPT14* homolog *PIG-A*. a** Cytotoxicity assessment by dose-response testing of jawsamycin and the control compound benzalkonium chloride against human HCT116 cells. **b** Genetic validation of the FLAER assay: genetic editing of the *PIG-A* gene leads to loss of extracellular concentration of GPI-anchored proteins in HCT116 cells as assessed by staining with the fluorescently labeled GPI-binding compound aerolysin (FLAER) and flow-cytometry analysis. **c** Incubation of HCT116 cells with different doses of jawsamycin has no apparent effect on GPI-anchored proteins as observed by FLAER staining and flow-cytometry analysis. For each histogram in panel **a** and **b**, 10,000 events were recorded and peak heights normalized. Unstained and untreated control samples were used to set the gating strategy (gating strategy images and additional data are provided in the Source Data file).

identified to inhibit Spt14 and thus the first step in GPI biosynthesis, we expected a fungicidal effect upon compound exposure. We incubated spores or germlings of *R. oryzae* with DMSO or 0.03 μM of jawsamycin followed by staining with propidium iodide. Propidium iodide weakly stains cell walls of intact fungal cells but strongly intercalates into DNA when the membrane integrity is compromised. When analyzing the result by fluorescence microscopy, we observed pronounced staining and apparent cellular swelling and leakage in compound treated samples in the majority of the cells (Fig. 2g). This observation is in line with jawsamycin inhibiting GPI biosynthesis and consequently compromising cell viability and demonstrating a fungicidal effect.

**Jawsamycin does not inhibit PIG-A the human Spt14 homolog**. The Spt14 enzyme is also conserved in mammalian cells, named PIG-A, with an overall identity ~40% compared to the fungal homologs. To assess if jawsamycin also inhibits the mammalian enzyme, we tested cytotoxicity on human HCT116 cells. No effect on the mammalian cell viability could be observed up to the maximal tested dose of 10 μM (Fig. 3a). Testing was extended to HEK293, HEPG2, and K562 cell lines using the WST-1 cytotoxicity assay but no effect could be detected up to 50 μM. Although some mutations in *PIG-A* are reported to be embryonic lethal in mice[27], human cells with *PIG-A* mutations are reported to be viable but lead to paroxysmal nocturnal hemoglobinuria (PNH) caused by exposure of blood cells to the complement system[28]. A standard cytotoxicity assay might thus not capture PIG-A inhibition in human cells. To directly monitor effects of jawsamycin on human PIG-A, we took advantage of the fluorescein-labeled proaerolysin (FLAER) test that is used to diagnose PNH[29]. FLAER binds selectively to the glycophosphatidylinositol anchor. Using a flow cytometry approach we benchmarked the decrease of FLAER positive cells when exposed

to jawsamycin. Specifically, genetic inactivation of *PIG-A* using the CRISPR-Cas9 system lead to a strong increase in FLAER negative cells, whereas no such effect was observed with the jawsamycin-treated cells over a period of 6 days (Fig. 3b, c). Thus, we conclude that jawsamycin is a potent antifungal lead compound with good selectivity for the fungal enzymes.

**A rapid and versatile chemical derivatization protocol**. Jawsamycin is a unique and highly complex natural product containing five cyclopropyl moieties in the fatty acid tail. Its biosynthesis has been explored[20] and there have been numerous approaches to chemically synthesize the various parts of the molecule (reviewed by Pietruszka et al.[30]). The current routes described for the total synthesis of jawsamycin are lengthy and tedious and the limited synthetic scope for derivatizing jawsamycin hampered the accessibility to a diverse set of testable compounds. As an alternate approach we started with the natural compound isolated by the producer strain and developed protocols to make it amenable to rapid and versatile chemical derivatization.

To allow for variations of the 5′-amino-5′-deoxy-5,6-dihydrouridine nucleoside moiety we optimized the hydrolysis of jawsamycin to obtain the intact cyclopranated carboxylic acid (compound 1, see Supplementary Methods). This part of the molecule subsequently served as starting material for the coupling of a limited series of nucleosides in order to explore antifungal structure-activity relationships (Fig. 4). In series a, only minor modifications of the dihydrouracil moiety were tolerated leading to reduced potency and antifungal activity against the four selected strains. The uridine analog JD-4 already loses activity on two strains, replacement of a ketone by a primary amine or introduction of a methyl group abolishes antifungal activity, as shown for compounds JD-2 and JD-3. In line with the identification that the compound inhibits the UDP-glycosyltransferase Spt14, exchange of the headgroup by purine

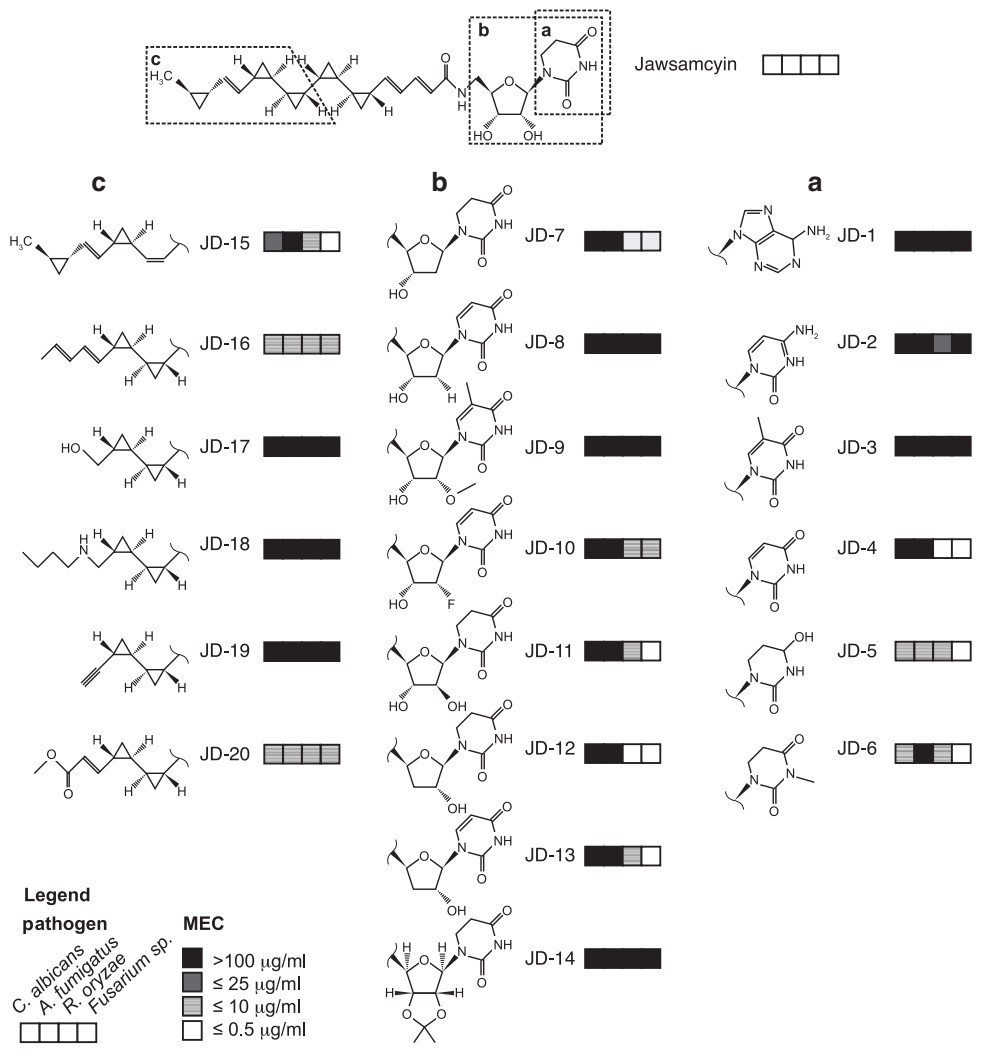

**Fig. 4 Chemical derivatization and bioactivity profiling of jawsamycin.** Chemical variation of the dihydro uracil moiety (**a**-series) the dihydro uridine moiety (**b**-series) and the cyclopropylated fatty acid tail moiety (**c**-series) was undertaken with the natural compound as starting material. The obtained jawsamycin derivatives (numbered JD-X) were purified, structure verified and tested against the indicated four different fungal pathogens to obtain minimal effective concentrations (MEC).

rendered compound JD-1 inactive. The *N*-methyl analog JD-6 retained activity for three fungal strains but not for *A. fumigatus*. Any elimination or substitution of the two hydroxy ribose moieties resulted in a significant drop of potency as exemplified with compounds JD-7 - JD 14 of series b. Although not comprehensive, this structure-activity analysis suggests that any deviation from the natural uridine substrate is not well tolerated for inhibition of the UDP-glycosyltransferase.

Modifications of the cyclopropanated fatty acid moiety were undertaken with the acetonide protected jawsamycin JD-14. This allowed modifications at positions C14 and C16 in the fatty acid tail (series c, Fig. 4), introducing functional groups such as hydroxy, butylamine, ethinyl and vinyl methylester (JD-17–20). JD-16 bearing a double bond in position 16/17[31] was one of the few analogs isolated from the fermentation broth. It showed decent potency against the four fungal strains. As for modifications of the headgroup, also derivatization of the fatty acid tail resulted in dramatic drop in activity, except for the methylester analog JD-20 which revealed an antifungal profile similar to jawsamycin (Fig. 4).

Despite the observation that none of the tested modifications displayed improved potency or spectrum compared to the parent compound, we provide a robust and rapid synthetic protocol for the generation of extended libraries of jawsamycin analogs which allows rapid screening for antifungal activity.

**Jawsamycin is efficacious in a pulmonary mucormycosis mouse model.** As the limited chemical derivatization described above did not identify a compound with improved potency, we tested the antifungal potential of the original jawsamycin compound in vivo. The observed in vitro potency against Mucorales fungi led us to examine the efficacy of the compound in an in vivo model of invasive pulmonary mucormycosis due to intratracheal instillation of *R. delemar*[32] (Fig. 5a). As depicted in Fig. 5b, c, oral dosing of jawsamycin at 100 mg/kg delivered for 6 consecutive days starting 24 h post infection, led to improved overall survival rate versus placebo-treated mice (45% and 10% overall survival for jawsamycin- and placebo-treated mice, respectively, *P* = 0.001). A lower dose of 30 mg/kg of jawsamycin or a dose of the clinically used posaconazole tended to improve survival compared to placebo-treated mice with *P* = 0.08 for both drugs. Administration of jawsamycin for 11 days did not improve overall survival over the 7-day treatment regimen. The enhanced survival in the jawsamycin-treated mice was corroborated by a significant ~1 log reduction in the fungal burden in both lung and brain when compared to placebo mice (Fig. 5d, e). Importantly, jawsamycin

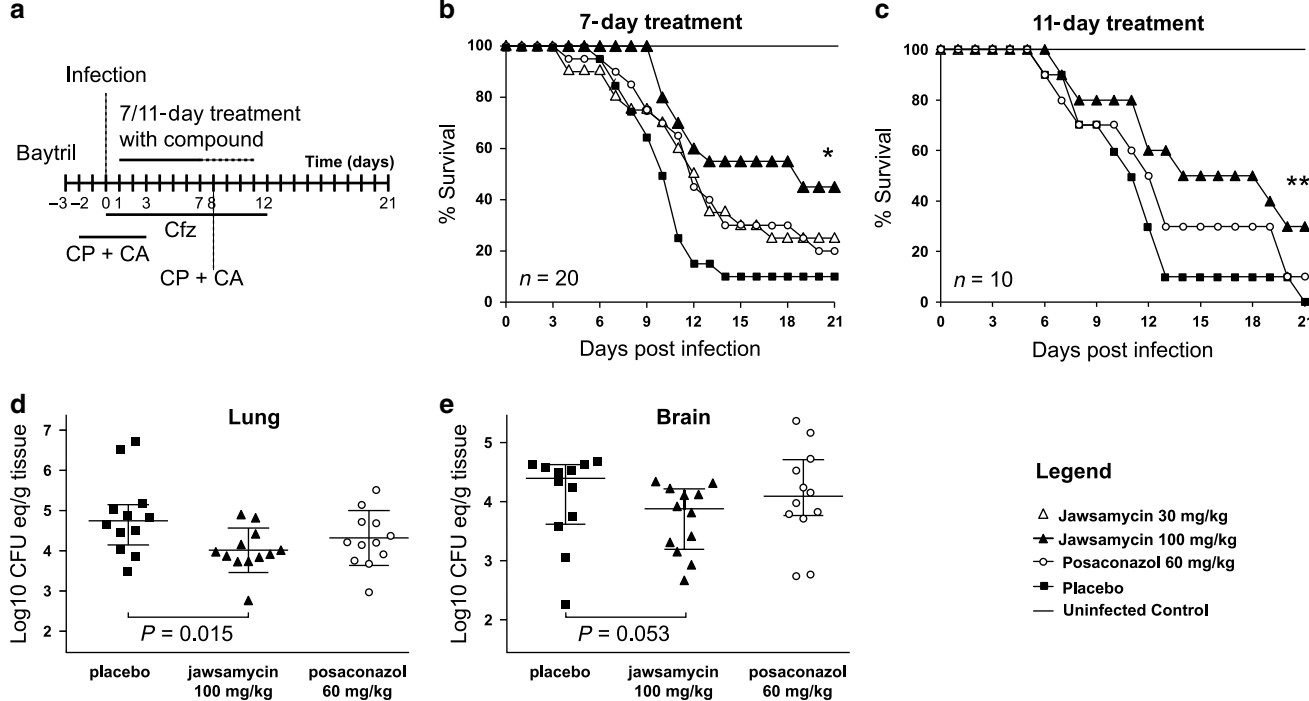

**Fig. 5 Testing of jawsamycin in a murine pulmonary mucormycosis model. a** Outline of experiment. On day 0, mice were infected with $2.5 \times 10^5$ spores of *R. delemar* by intratracheal instillation. CP cyclophosphamide, CA cortisone acetate, Cfz ceftazidime. Mice were treated for either 7 days (**b**) or 11 days (**c**) with the indicated compound treatments. **b, c** Kaplan–Meier plots depicting survival of mice under the different treatment regimens over 21 days. **b** $n = 20$ mice/group from two independent experiments with similar results (confirmed average fungal spores delivered to lungs of $2.4 \times 10^3$). **c** $n = 10$ mice/group from one experiment (confirmed fungal spores delivered to the lungs of $3.7 \times 10^3$). *$P = 0.0014$, and **$P = 0.045$ vs. placebo in **b**, **c**, respectively. **d, e** Tissue fungal burden of lungs and brain harvested from 12 mice/group and processed on day 7 post infection. Tissue fungal burden was determined by real-time qPCR and expressed as Log10 CFU (colony forming units) equivalent/gram of tissue. *P* values < 0.05 were considered significant (details on data analysis and statistical significance are described in the Methods section).

appeared to be well tolerated by mice with no signs of toxicity at the tested doses.

In summary, this is the first evidence that targeting fungal Spt14/Gpi3 by jawsamycin results in a beneficial effect against pulmonary *Rhizopus* infections. The demonstration of in vitro efficacy, the broad-spectrum activity against fungi in vitro, and its novel mechanism of action, warrants further investigations of jawsamycin as a novel antifungal lead.

## Discussion

The GPI pathway has previously been proposed as a promising clinical intervention point to treat microbial infections[9,12,14]. However, therapeutic exploitation has been hampered by the availability of few chemotypes targeting only two nodes[5,6,10]. Here, we introduce a high-throughput screen which aimed at the identification of novel GPI biosynthesis inhibitors which led to the identification of the unique natural product jawsamycin (FR-900848). Genetic follow up pointed towards the phosphatidyli-nositol *N*-acetylglucosaminyltransferase complex (Gpi1, Gpi2/, Gpi15, and Spt14/Gpi3) as the affected node of the pathway. Furthermore, unbiased resistance profiling studies pinpointed the catalytic subunit Spt14 as the target of jawsamycin. It is unfortunate that no crystal structure of Spt14 was available at the time this study was released. Mapping of resistance conferring residues into the protein structure would likely allow us to identify the binding pocket of the compound as accomplished in other studies and thereby support structure guided design of analogs[33]. The closest functionally related proteins for which crystal structures are available are the *E. coli* UDP-N-acetylglucosamine epimerase (1VGV)[34] and phosphatidyl mannosyltransferase from *Mycobacterium smegmatis* (4N9W)[35]. However Spt14 homology is low (~20% identity) and does not allow for calculation of homology models. Despite this, it is obvious that the headgroup of jawsamycin is a close analog of uridine thus it is plausible to hypothesize that jawsamycin engages at the active site and occupies the UDP binding pocket. This notion was supported by our chemistry efforts, in which the headgroup tolerated only minimal changes without complete loss of activity and thus only tolerated alterations still resembling the dihydro-UDP pharma-cophore. One caveat of direct analog testing on fungi was that it did not allow differentiation between altered activity due to compound efflux or activity on the target. Although total synthesis of jawsamycin has been described[35,36], our optimized protocols for fermentation/isolation and chemical derivatization of the natural compound provide a valuable starting point to expand medicinal chemistry programs to further optimize the lead compound jawsamycin.

Our chemogenomic HOP profiling has identified the UPR key node Hac1/Ire1[36,37] to be synthetic lethal with inhibition by jawsamycin. This is in agreement with the notion that non-glypiated proteins are retained in the ER and consequently dis-rupt proteostasis[5]. Thus, it appears that inhibition at the level of Spt14 not only inhibits an essential function but furthermore exerts a dominant negative function. It has been demonstrated that compromising GPI biosynthesis result in cell-wall defects. Again, this is reflected in the high quality HIP HOP profiles where the two genes *JIP3/MID2* score prominently in HOP. *JIP3* encodes a dubious ORF and almost entirely overlaps the verified *MID2* gene. Thus, it is reasonable to consider this as a second *mid2* deletion strain. Mid2 acts as a sensor for cell wall integrity and activates the integrity signaling pathway. Thus, the

chemogenomic profiling studies have identified the key relevant aspects of GPI biosynthesis: the targeted UDP-GlcNAc phosphatidylinositol transfer complex in HIP and the downstream consequences of target inhibition, UPR, and cell wall defects, in HOP.

Interestingly, the *MUD2* gene coding for a protein involved in early pre-mRNA splicing significantly scored in the HOP profile of jawsamycin but not in those of the Gwt1 inhibitor CMB2683 a close E1210 derivative, nor the Mcd4 inhibitor M743/CMB1539[22]. As *SPT14* is one of few genes in *S. crevisiae* that contains an intron affecting *SPT14* mRNA maturation in additon to inhibition of the protein by jawsamycin exerts synthetic lethality. This can be considered yet another line of support that *SPT14* encodes the primary target of jawsamycin.

The mechanism of action was also apparent when performing microscopic analysis of *Rhizopus* spores upon incubation with jawsamycin. The compound treatment led to profound spore swelling, with the diameter range 15–20 µm, frequent leakage of cellular content and consequent cell death. This is consistent with defect in the integrity of fungal cell wall, whose biosynthesis requires many GPI-anchored proteins.

Recent positive data with the clinical Gwt1 inhibitors of the APX001 class in models of pulmonary aspergillosis, coccidioidomycosis, and against *Candida auris* validate the GPI pathway as a valuable druggable biological intervention point to tackle fungal diseases[15,38,39]. With the availability of only a few chemotypes which modulate just two targets (Gwt1 and Mcd4) it is apparent that this pathway is underexplored and merits more attention. It is to be hoped that the currently available compounds, including jawsamycin, will be used as chemical starting points for additional screening campaigns.

In summary, jawsamycin is the first identified inhibitor for the fungal UDP-GlcNAc phosphatidylinositol transfer complex, shows selectivity over the mammalian enzyme, and exerts antifungal properties against a wide-spectrum of pathogenic fungi in vitro, and an intractable pathogen in a murine disease model. Combined with a robust protocol to chemically modify both, the headgroup and the polycyclopropyl tail, our studies provide a possible starting point for antifungal drug development.

## Methods

**GPI-reporter strain construction**. The DNA sequence encoding *Gaussia princeps* luciferase fused to GPI anchor as described by Enjalbert et al.[18] was synthesized (Genewiz) and cloned into EcoRI/BglII sites of pESC-URA plasmid. pGluc59, the resulting plasmid was transformed into YPH499 or NF7061 for characterization of luciferase expression and GPI-reporter assay development.

**GPI-reporter assay development**. For assay development, yeast cells (NF7061 transformed with pGluc59) of an overnight culture grown in 20 ml glucose selection medium at 30 °C under shaking at 215 rpm, were pelleted and resuspended to $2 \times 10^7$ cells/ml in 50 ml glucose selection medium. After an additional incubation of 8 h, cells were resuspended in 150 ml raffinose selection medium at a concentration of $1 \times 10^7$ cells/ml, and incubated for additional overnight incubation at 30 °C under shaking at 215 rpm. Afterwards cells were pelleted and resuspended in raffinose rich medium to a final concentration of $12 \times 10^7$ cells/ml.

10 nl/well of a 10 mM compound stock solution were spotted into assayplates (1536-well, black, µclear with lid 190 µm, Greiner #792091-191) using acoustic dispensing (Echo 550, Labcyte) yielding a final compound concentration of 17 µM. Then 3 µl of galactose induction medium, followed by 3 µl of the yeast suspension of $12 \times 10^7$ cells/ml are added to the sample wells using a Multidrop Combi device (Thermo Fisher). After incubation for 16 h, 4 µl of the supernatant from the primary assay plate were transferred into a secondary assay plate using the FDSS 7000 (Hamamatsu Inc.) 1536-well pipetting head. Next 2 µl of luciferase reaction mixture (30 µM coelenterazine dissolved in luciferase reaction buffer (150 mM $KH_2PO_4$, 750 mM NaCl, 1.5 mM EDTA in ultra pure $H_2O$, pH 6.7, sterilized by filtration) was added, and imaged using the FDSS 7000 reader in luminescence mode and a reading time of 30 s.

**GPI-reporter gene screen**. For screening of inhibitors of the GPI pathway, 20 ml of glucose selection media (yeast nitrogen base without AA 6.7 g/l; D-glucose 20 g/l;

CSM-URA 0.8 g/l) was inoculated with the yeast strain NF7061 (BY4741 MATa with snq2::KanMX; pdr5::KanMX; pdr1::NAT1; pdr3::KanMX; yap1::NAT1; pdr2::LEU2; yrm1::MET; and yor1::LYS2) transformed with pGluc59 and incubated at 30 °C under shaking for 24 h. Cells were harvested and resuspended to $2 \times 10^7$ with 50 ml glucose selection media and grown for additional 8 h. Cells were counted and raffinose selection media (yeast nitrogen base without amino acids 6.7 g/l; D-raffinose 20 g/l; and CSM-URA 0.8 g/l) was inoculated at a cell concentration of $1 \times 10^7$ cells/ml. After 16 h incubation under shaking and at 30 °C, cells were resuspended to a concentration of $1 \times 10^7$ cells/ml in raffinose rich media (D-raffinose 20 g/l; peptone 20 g/l; yeast extract 10 g/l and CSM-URA 0.8 g/l) and further incubated for 60 min at 30 °C under shaking. In all, 1536-well assay plates (white, solid bottom, Greiner # 792184-191) were prepared by addition of 30 nl of a 2-mM solution of compounds (yielding a final concentration of 10 µM), to lanes 1–44 using acoustic dispensing (Echo 555, Labcyte Inc.). The neutral control (DMSO) was placed within lanes 45 and 46. Afterwards 3 µl of galactose induction media (D-galactose 40 g/l; peptone 20 g/l; yeast extract 10 g/l; and CSM-URA 0.8 g/l) was added to lanes 1–46 using a Multidrop combi dispenser (Thermo Inc.). In total, 3 µl of galactose induction media supplemented with the positive control compounds are added to lanes 47–48 with the help of a Preddator dispenser (Redd and Whyte Inc.). In all, 3 µl of yeast suspension ($1 \times 10^7$ cells/ml) are added to each well of the 1536-well plate using a Multidrop combi (Thermo). After an incubation at 30 °C for 8 h, 4 µl of the supernatant of each well was transferred into an additional assay plate, mixed with 2 µl of a luciferase reaction mixture and incubated for 20 min at room temperature. For luciferase reading a Pherastar FS reader (BMG Labtech, reading time 0.1 s; focus height 0.8 mm; gain 4000) was used.

The data was analyzed with the internally developed Helios software tool[40] and normalized to the controls (DMSO negative control, a Gwt1 inhibitor at a final concentration of 10 µM corresponding to its $IC_{50}$ value was used as active control, using the following calculation:

$$\text{NC1:} \quad xn = +/- 100\,(x - NC)/(AC - NC)$$

±effect type parameter; NC and AC are averages (median) over the corresponding neutral control (NC) and active control (AC) values.

**Growth inhibiton testing**. Substances were first assayed for their potency using wild-type *S. cerevisiae* BY4743 by recording growth inhibition dose-response curves. $OD_{600\ nm}$ values of exponentially growing *S. cerevisiae* cultures in rich medium with a robotic system. Twelve point serial dilutions, with a dilution factor of 3.16, were assayed in 96-well plates with a reaction volume of 150 µl. Cell growth was recorded as optical density at 600 nm using a Beckman DTX880 plate reader with Multimode detection software v3.3.09. Solutions containing dimethyl sulfoxide (DMSO) were normalized to 2%. $IC_{50}$ values were calculated using $n = 2$ data and logistic regression curve fits generated by GraphPad Prism (GraphPad Software, San Diego, California USA).

**Chemogenomic profiling**. The growth-inhibitory potency of test substances was determined using wild-type *S. cerevisiae* BY4743. $OD_{600}$ values of exponentially growing *S. cerevisiae* cultures in rich medium were recorded with a robotic system and optical density measured at OD 600 on a Beckman DTX880 plate reader with Multimode detection software v3.3.09. Twelve point serial dilutions were assayed in 96-well plates with a reaction volume of 150 µl. Solutions containing dimethyl sulfoxide (DMSO) were normalized to 2%. $IC_{30}$ values were calculated using logistic regression curve fits generated by TIBCO Spotfire Spotfire 7.9–10.3 (TIBCO Software Inc).

The *Saccharomyces* haploinsufficiency profiling (HIP) and homozygous profiling (HOP) and microarray analysis was performed as published previously[22]. The basic concept behind this assay is that HIP identifies genes where one functional copy, compared to two, confers hypersensitivity to inhibition by the compound. This indicates pathways directly affected by the compound. HOP (both gene copies deleted) indicates synthetic lethality and identifies compensating pathways to those directly affected by the compound. Thus genome-wide hetero- and homozygous deletion libraries of *S. cerevisiae* strains were purchased, (OpenBiosystems, Cat #YSC1056 and YSC1055) and pools were constructed as published previously[22]. Each HIP strain is heterozygous and each HOP strain completely null for one gene (and each strain being identified by a unique DNA sequence, called "bar-code" or "tag" inserted into the deleted gene). For each HIP and HOP experiment, each test substance was assayed in duplicate (2 wells) at its $IC_{30}$ concentration, in 24-plates (Greiner 662102), with 1600 µll/well YPD. DMSO was normalized to 2%. At the onset of the HIP experiment ~250 yeast cells/strain (100 µl of a 1.5 $OD_{600}$/ml culture) from an overnight log phase pre-culture where added to the compound and DMSO containing wells. Plates were incubated for 16 h in a robotic shaking incubator at 30 °C/550 RPM allowing for ~5 doublings. In all, ~250 yeast cells/strain (120 µl of a 1.2 $OD_{600}$/ml culture) were subsequently transferred into a new 24-well plate. Once inoculated, the new plate was incubated at 30 °C/550 RPM to allow the next 5 yeast generations (generation 6–10). This procedure was repeated 2x more until the final plate containing the yeast with ~20 generations were stored at 4 °C. The HOP assay was performed similar to the HIP experiment but the duration was reduced to 16 h/~5 doublings and thus no back-dilutions were necessary. Before the experiment, aliquots of the HOP pool were thawed and recovered for 3 h in YPD, then ~320 yeast cells/strain

(110 µl of a 1.5 OD$_{600}$/ml culture) were transferred into each well. The cell material from the final HIP and HOP plates was harvested, the gDNA extracted and the tags amplified as published previously[22]. Then relative abundance of each strain in the compound treated wells was compared to eight no-drug control samples that were produced in the same experiment.

For the experimental analysis we used the same computation of normalized tag intensities, outlier masking, and saturation correction as published previously[22,41]. Sensitivity was computed as median absolute deviation logarithmic (MADL) score for each compound/concentration combination, then gene-wise z-scores based on a robust parametric estimation of gene variability from >3000 different profiles allowing for up to 15% outliers were computed as described in detail[22]. Profiles were visualized using TIBCO Spotfire 7.9–10.3.

**Selection of drug-resistant _S. cerevisiae_.** Strain BY4743Δ8, derived from BY4741 but deleted for eight genes involved in drug resistance (efflux pumps: _SNQ2_, _PDR5_, and _YOR1_; transcription factors: _PDR1_, _PDR2_, _PDR3_, _YAP1_, and _YRM1_) was incubated in 2.5% ethylmethane sulfonate until only 50% of the cells formed colonies. A total of 2 × 10$^7$ mutagenized cells were plated on two 14-cm dishes with synthetic complete medium (0.7 g/l Difco yeast nitrogen base w/o amino acids, 0.79 g/l MPbio CSM amino acid mixture, and 2% Glucose) containing 500 nM jawsamycin. After 4 days resistant colonies could be isolated and resistance confirmed by restreaking on selective medium. Genomic DNA extraction, whole-genome sequencing and focused sequencing to confirm findings were performed as published[22]. Mutations were cloned into non-mutagenized wild-type cells and fold resistance compared to wild-type assessed by dose-response testing.

**Mammalian cell cytotoxicity testing.** Effects on mammalian HCT116 cells, (CCL-247, ATCC) were assayed by testing serially diluted compounds in 384-well plates seeded with 750 cells/well in DMEM high glucose, Glutamax, pyruvate medium (31966047, Life Technologies). DMSO was normalized to 0.2%. Viability was read by following the CellTiter-Glo protocol (G7573, Promega) after 96 h compound treatment.

A 50% benzalkonium chloride solution (63249, Sigma Aldrich) was used as cytotoxic, positive, and full inhibition control. This reagent was chosen as it is inexpensive, stable, results in a steep inhibition curve, and does not interfere with a large range of readouts and safe for handling. The WST-1 cytotoxicity/proliferation assay was performed according to manufacturer's instructions (Roche Cat # 11 644 807 001) to test compounds for growth inhibition over 72 h in 3 mammalian cell lines: HEK293 (CRL11268, ATCC), K562 (CCL-243,ATCC), and HEPG2 (HB-8065, ATCC). Effects on PIG-A and GPI-anchored proteins were tested by the FLAER assay[29] as follows: 50,000 cells/well were seeded in a 6-well plate and incubated the following day ±jawsamycin. After 4 days treatment, the cells were split and stained using a 1:100 dilution of the FLAER reagent (FLAER Alexa Fluor 488 proaerolysin, FL2S, Cedarlanelabs) in PBS + 3%BSA for 20 min. For genetic perturbation of _PIG-A_ we used a stably expressing Cas9 HCT116 cell line and transduced it with a lentiviral construct encoding for the sgRNA. The crRNA part contained the following sequence against _PIG-A_: TGGCGTGGAAGAGAGCATCA and as control a sgRNA containing a non-targeting crRNA sequence was used: GTAGCGAACGTGTCCGGCGT. To allow for editing and protein depletion cells were analyzed 11 days after transduction and stained as described above. The cells were analyzed on a BD FACSAria Fusion using FACSDiva 8.0.1 and FlowJo 10.1r1. 10,000 events were recorded. For jawsamycin effect testing, untreated, and unstained cells were used to set the gates for negative events. Untreated, FLAER-stained cells were used to set the gates for positive events. In the genetic control experiment, FLAER-stained cells with the non-targeting sgRNA were used to set gates for positive events and unstained cells were used as a negative control. Raw data and gate setting images are provided in the Source Data file.

**Antifungal testing.** Antifungal susceptibility testing was performed according to the Clinical and Laboratory Standards Institute (CLSI) guidelines for broth microdilution M27-A3 (yeast) and M38-A2 (mold) in RPMI 1640 (HyClone; SH30011.03) with 2.05 mM glutamine and phenol red, without bicarbonate, and buffered with 0.165 M MOPS [3-(N-morpholino)propanesulfonic acid] (AppliChem; A1076,0250) to pH 7.0.

**Producer strains and biosynthetic gene cluster.** The jawsamycin (FR-900848) sample in the screening deck was annotated to be derived from a _Streptomyces luteoverticillatus_ strain in our strain collection. Our records describe that the strain was isolated from a soil sample in China, Wuhan province and added to the Novartis collection in the context of a collaboration in 2004. Ilumina sequencing of _Streptomyces luteoverticillatus_ revealed a total genome size of 7.84 Mbp. BLAST search allowed to identify the biosynthetic gene cluster (BGC) of jawsamycin. The published jawsamycin BGC (accession no. AB920328) shows extremely high similarity at the amino acid level upon comparison to the sequence obtained from our internal strain (average amino acid identity of 99.6%)[20]. The obtained sequence of the biosynthetic gene cluster including some flanking sequence can be found in the Source Data file and has been deposited in the GenBank nucleotide database with accession number MT553334. This strain was subsequently used for further sourcing of jawsamycin as described in the following sections.

**Media optimization and fermentation.** The producer strain was cultivated at 28 °C for 7 days under agitation in shake flasks and the titers of jawsamycin were quantified by UPLC-UV. Medium 300-04.00, consisting of agar (Bacto) 1 g/l, glycerol 1 g/l, NaCl 0.05 g/l, CaCO$_3$ 0.05 g/l, KH$_2$PO$_4$ 0.25 g/l, K$_2$HPO$_4$ 0.5 g/l, MgSO$_4$ 0.1 g/l, yeast extract 1.35 g/l, NZ-amines 2.5 g/l, malt extract 5.85 g/l, L-asparagine monohydrate 1 g/l, soy protein (Unico 75) 2.5 g/l, potato starch (Noredux A150) 7.5 g/l, cerelose 7.5 g/l, HEPES 6 g/l, trace element stock solution 1 ml/l), and with a titer of 2.2 mg/l jawsamycin was selected for initial fermentations and as starting point for media optimization. For the optimization of the production medium a Plackett-Burman experimental design was applied as first measure in order to identify the media components that were most crucial for production of jawsamycin. The highest positive influence of media components was observed for potato starch, malt extract, yeast extract, and KH$_2$PO$_4$. These components were selected for further optimization in a central composite design experiment. In total, 25 different variations were prepared and used for cultivation followed by quantification of jawsamycin. The optimized medium 300–30.00, consisting of agar (Bacto) 1 g/l, glycerol 7.5 g/l, NaCl 0.05 g/l, CaCO$_3$ 0.05 g/l, KH$_2$PO$_4$ 0.5 g/l, K$_2$HPO$_4$ 1 g/l, MgSO$_4$ 0.1 g/l, yeast extract 15 g/l, NZ-amines 2.5 g/l, malt extract 15 g/l, L-asparagine monohydrate 1 g/l, soy protein (Unico 75) 2.5 g/l, potato starch (Noredux A150) 20 g/l, and cerelose 7.5 g/l, HEPES 6 g/l), and trace element stock solution 1 ml/l, resulted in a titer of 10.3 mg/l jawsamycin in shake flasks. The same medium, but without agar, was used for fermentations at the 3700 liter scale. In all, 3500 liters of medium were inoculated with 200 liters of seed culture and cultivated for 5 days at 28 °C. The pH was controlled at pH 7 ± 0.2 with H$_2$SO$_4$ and NaOH, and the pO$_2$ was controlled at 40% of saturation by means of the stirrer speed. After 5 days the fermentations were harvested with an average titer of 4–5 mg/l.

**Extraction and isolation.** The fermentation broths were adjusted to pH 5.5 at time of harvest and extracted with equal volumes of ethyl acetate. Phases were separated with a Westphalia SB21 separator. The organic layer was evaporated and the viscous oil obtained thereby was suspended in a mixture of methanol/water 9:1 and extracted with cyclohexane in order to remove lipids. The cyclohexane was discarded and the methanol/water phase was extracted two more times with cyclohexane. Methanol was evaporated and the aqueous layer extracted with ethyl acetate. The amounts of extracts obtained by this procedure from the large scale fermentations were in the range of 900 g–1 kg. The crude extracts were separated by successive chromatographic steps. Initial separation was carried out on large scale preparative HPLC (RP18 YMC ODSA SP, 5–15 µm, 200 × 700 mm). The mobile phases used were water and methanol, both supplemented with 0.1% formic acid. The flowrate was 500 ml/min and the gradient started at 50% methanol kept constant for 10 min, followed by a gradient step to 65% methanol and isocratic elution for 50 min. This was followed by another gradient step to 75% methanol and further isocratic elution until the separation was finished and the column cleaned with 100% methanol. During the preparation of the first sample for injection in 50% methanol and 50% water, a precipitate was observed, which turned out to consist mostly of jawsamycin. Therefore the crude extract was dissolved in 50% methanol and 50% water, concentrated by evaporation and stored at 4 °C overnight. The resulting precipitate was obtained by filtration and resulted in a significant enrichment of jawsamycin which was submitted to separation as described above. Final purification was performed by preparative HPLC on a Sunfire-C18 column (5 µm, 30 × 150 mm, Waters). The mobile phases used were water and acetonitrile, both supplemented with 0.1% formic acid. The flowrate was 60 ml/min and the gradient started at 50% acetonitrile kept constant for 1.5 min, followed by a linear gradient from 1.5 to 17 min from 50 to 70% acetonitrile, followed by flushing and reequilibration of the column. Fractions containing the compound were concentrated under reduced pressure, followed by freezing and lyophilization. Depending on the intended purpose of the material (starting material for semi-synthesis, in vitro studies or in vivo studies) jawsamycin was purified to 85–99% purity as assessed by UPLC-UV-CAD-MS and NMR.

In total several fermentations at volumes from 20 to 100 l and 3 fermentations at 3700 l scale were carried out and a combined 33.2 g of jawsamycin were purified.

**Chemical derivatization of jawsamycin.** For detailed protocols on the derivatization of jawsamycin, see the Supplementary Methods section in the Supplementary Data document provided as an online supplement.

**Mucormycosis mouse model and antifungal efficacy testing.** The mucormycosis model was conducted as previously described[32]. Male ICR mice (20–25 g) were purchased from Taconic (Germantown, NY) and were housed in groups of five mice each. They were given irradiated feed and sterile water containing 50 µg/ml Baytril (Bayer) ad libitum. Mice were immunosuppressed with cyclophosphamide (200 mg/kg administered i.p.) and cortisone acetate (500 mg/kg administered SQ) on days −2, +3, and +8 relative to infection. This treatment regimen results in ~16 days of leukopenia, with total white blood cell count dropping from ~13,000 cm$^3$ to almost no detectable leukocytes as determined by a Unopette system (Becton-Dickinson and Co.). To prevent bacterial infection, 50 µg/ml enrofloxacin (Baytril; Bayer, Leverkusen, Germany) was added to the drinking water from day −3 to day 0. Ceftazidine (Cfz, 5 µg/dose/0.2 ml) replaced

enrofloxacin treatment on day 0 and was administered daily by subcutaneous injection from day 0 until day 6 (tissue fungal burden) or day 13 (survival).

Intratracheal instillation of *R. delemar* spores was carried out by a modification of the method of Luo and coworkers[32]. Briefly, while pulling the tongue of isoflurane gas-anesthetized mice anteriorly to the side with forceps, 25 μl of fungal spores ($2.5 \times 10^5$ cells) in PBS was injected through the vocal cords into the trachea with a Fisherbrand gel-loading tip (catalog number 02-707-138). To confirm the inoculum delivered to the lungs, two or three mice were killed immediately after the infection, and lungs were dissected, homogenized, and quantitatively cultured on PDA plates plus 0.1% Triton. Colony forming units (CFU) were enumerated after incubation at 37 °C for 24 h. After infection, the mice were randomly sorted into different treatment groups. For uninfected control mice, 25 μl of PBS alone was intratracheally injected.

For testing jawsamycin antifungal efficacy, the compound was formulated as nanoparticle suspension in 1% hydroxypropyl cellulose (HPC)/0.1% Tween 80, and was given by oral gavage at daily dose of 30 or 100 mg/kg once daily (QD). As an active comparator arm, posaconazole oral suspension was given by oral gavage at a daily dose of 60 mg/kg QD. Infected untreated control mice (placebo) received 1% HPC/0.1% Tween 80. The therapies, were initiated 16 h after the fungal infection, and were given for 7 or 11 d. Time to morbidity with moribund mice humanely euthanized served as endpoint point and followed for 21 d post infection.

In separate experiments treatment with the drugs started after 16 h post infection as above and continued until Day +6 post infection. Mice were killed on Day +7, and the lungs and brain were excised and processed for tissue fungal burden determination by real-time qPCR as previously described[32]. Data of tissue fungal burden were presented as Log10 CFU equivalent/gram of tissue.

For survival studies, 10 mice/group would provide at least 80% power to test the hazard ratio of 0.2 or more with a level of significance $p = 0.025$ using Log Rank test and Cox proportional model (one-sided test) assuming 100% and 50% mortality in the control and treated groups, respectively. For the tissue pathogen burden, 10 mice/group would provide at least 90% power to detect the effect size of 3 or 3 SD difference in CFU (expressed as log) by two sample $t$-test with $\alpha$ of 0.05, assuming the standard deviation of the treated group is twice of the one for the control group. A two sample $t$-test and ANOVA were used with the post-hoc analysis using Tukey correction methods to control for the overall type error rate of 0.05. For all comparisons, median (Interquartile range), and 95% confidence interval were computed. All data analyses were conducted using GraphPad Prism 6. $p < 0.05$ were considered significant.

All animal related study procedures were compliant with the Animal Welfare Act, the Guide for the Care and Use of Laboratory Animals, and the Office of Laboratory Animal Welfare and were conducted under an IACUC approved protocol by the Lundquist Institute at Harbor-UCLA Medical Center.

**Reporting summary**. Further information on research design is available in the Nature Research Reporting Summary linked to this article.

## Data availability

The source data underlying Figs. 1b, d–g, 2a, b, e, and 3a–c and Supplementary Figs. 1 and 2 are provided in the Source Data file. The synthesis routes and analytical spectra of the chemical derivatives presented in Fig. 4 are provided in the Supplementary Information document. Chemical structures in ChemDraw format can be found in the Source Data file. The nucleotide sequence of the jawsamycin biosynthetic gene cluster has been deposited in the GenBank database with accession code MT553334. Source data are provided with this paper.

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

## Acknowledgements

We thank the Novartis Natural Products and Biomolecular Chemistry team for high quality compound supply, Nicole Hartmann and Jürg Eichenberger for processing of the HIP HOP microarrays and Susan Kirkland for proofreading the manuscript. The help of Teclegiorgis Gebremariam and Sondus Alkhazraj in the animal studies is greatly appreciated. The animal studies were supported by Public Health Service contract HHSN272201000038I (NIH task orders A13 and 93) and grant R01 AI063503.

## Author contributions

D.E., C.S., T.A., and R.R. performed the target identification experiments; Y.F., F.F., and F.G. designed and executed the GPI screen; Y.F. conducted fungicidal characterizations; D.P., K.M., E.W., E.R., and F.P. were involved in isolation and purification of the natural compound; synthesis of compound derivatives was performed by M.M., S.R., and J.H.; antifungal testing was conducted by V.P. and A.S.I.; bioinformatic analysis was done by S.S. The project was conceived, figures designed and the manuscript written by J.T. and D.H. with feedback from all authors.

## Competing interests

The authors declare no competing interests.
