## [Peer Review File · Nature Communications]

Reviewers' comments:

Reviewer #1 (Remarks to the Author):

Systemic fungal infections are responsible for high rates of mortality and are a serious public health challenge. Since they are eukaryotes like the host, it has been difficult to find drugs that specifically target the pathogen. Hence, a very limited repertoire of drugs is available to treat fungal infections. Also, as use of these antifungals have become widespread, the pathogens responsible for disease too have begun to shift to more drug resistant fungal species on which the current drugs have poor efficacy. Thus, the search for new, effective antifungals is urgent. The present manuscript studies the possibility of using Jawsamycin (a natural product identified during the course of this study as a fungal-specific GPI biosynthetic inhibitor) as a new antifungal with potential clinical applications.

The study is timely, well designed/executed and likely to be of interest to a wide audience. I would recommend publication but after major revisions.

The authors provide a reasonably good Introduction to the study and to the challenges of finding new fungal inhibitors/ antifungals. The results are potentially very exciting. However, there are several errors. The most serious of these is the fact that the figure legends either do not match with the text or the relevant figure or both.

The authors begin by performing a high-throughput screen of 12472 compounds from the Novartis library of natural products to test their effectiveness against the GPI anchor biosynthetic pathway in fungi. Their rationale for choosing this pathway as a drug target is quite convincing: targeting the GPI anchor biosynthesis should also subject the pathogen to cellular stress, compromise its cell wall integrity and expose the cell wall beta- glucans to the host's immune response pathway. There are also some differences between the mammalian and fungal GPI biosynthetic enzymes, which they hoped to be able to exploit to make the drug specific for the pathogen.

A quick method for screening the compounds was developed. For this purpose, the *G. princeps* luciferase gene was fused to the GPI anchoring signal sequence of *Candida albicans* PGA59 and attached to the inducible GAL1/10 promoter and expressed in *S. cerevisiae*. Known antifungals, voriconazole and amphotericin B (which do not target GPI biosynthesis), as well as the GPI inhibitor, E1210 (which inhibits the Gwt1 enzyme), were used to show that the assay methodology works and is specific for inhibition of GPI biosynthesis. Inhibition of GPI biosynthesis alone results in the luciferase enzyme being released into the medium rather than remaining bound to the cell surface. This was then used to screen the natural products library, which led to the identification of an oligocyclopropyl-containing natural product, named Jawsamycin. This was further validated by using a fresh batch of Jawsamycin purified from the producer strain, *Streptomyces luteoverticillatus*, and yielded an IC₅₀ of 7 M in the assay. Also, since Jawsamycin inhibited growth of *S. cerevisiae*, it was next used in HIP/HOP assays with *S. cerevisiae* to identify potential targets.

- The rationale for using 10 μM for voriconazole, and 1 μM for amphotericin B and E-1210 in the experiment needs to be explained (Figure 1b).
- In the text, IC₃₀ for growth inhibition by Jawsamycin is stated to be 800 nM. No details are given in the figure legends, but in Figure 1g, IC₅₀ is mentioned instead of IC₃₀. Which should it be? Also, details of the experiment and how growth was monitored need to be given either here or in the Methods section.
- What do the error bars in Figure 1 represent?
- Nomenclature is a problem at several places in the text. For example, is it *Streptomyces luteoverticillatus* or *Streptoverticillium fervens*?

Two independent HIP experiments identified genes of the enzyme complex involved in the first

step of GPI biosynthesis, GPI1, GPI2, GPI3 (SPT14) and GPI15, as possible targets with good z-scores. The growth inhibition of individual strains with Jawsamycin was retested for confirmation. Not surprisingly, genes of the unfolded protein response pathway (HAC1 and IRE1) and the cell wall integrity pathway (MID2/JIP3, RLM1) were synthetic lethal with GPI biosynthesis inhibition by Jawsamycin and showed up in HOP experiments.

- The experiments also identified the pre-mRNA splicing component, MUD2, as synthetically lethal with the genes of the GPI biosynthetic pathway (as seen in the figure). A discussion of its relevance is required.
- The data given in Supplementary Figure S1 does not mention the value of 'n'. Also, it is not clear whether IC50 or IC30 was tested. OD600 should be OD600 nm.
- The gene name, SPT14, should be replaced by GPI3 to keep the nomenclature uniform for the GPI genes.
- It is rather surprising that GPI19, which is an essential gene, did not get identified as a target. At least a mention of the fact that the enzyme complex has two additional subunits which were not identified in the HIP assay should be mentioned.

Using an unbiased mutagenesis screen, mutations were identified in SPT14 (heterozygous H106Y mutation, V76F mutation and T144A mutation) that provided resistance to Jawsamycin. No mutations were identified in GPI1, GPI2 or GPI15. To confirm that the mutations led to resistance, these were then introduced in the wild type SPT14 gene. A 10-100 fold shift in the IC50 confirmed that the mutations conferred resistance to Jawsamycin. The mutations had no effect on sensitivity to clotrimazole, an Erg11 inhibitor. Strong propidium iodide staining suggests that the action is cidal in nature.

- Figure 2c: The grey domain in the figure is mentioned as 'predicted PI-n-acetylglucosaminyltransferase subunit'. Firstly, this is not a subunit. Perhaps the authors meant to identify it as the predicted catalytic domain. Also, it should be PI-N-acetylglucosaminyltransferase rather than 'PI-n-acetylglucosaminyltransferase subunit'.
- Legends for Figure 2d state: 'd) Frequency, IC50 concentrations and fold resistance of identified mutant clones as calculated by using the curves depicted in panel E and F.' No panels E and F exist in the figure. It should be panel 'e' since only the data for Jawsamycin have been listed. The data from panel 'f' too should be given or the range summarized in the legends.
- Was PI staining of Jawsamycin-treated cells done for *A. fumigatus* or *R. oryzae*? The figure legends (Figure 2g) say that the data presented is for *A. fumigatus*, while the text mentions *R. oryzae*.
- PI staining of at least one species from Mold and one from Mucorales would have been useful, given that the cell wall architectures are not the same in these cases.
- Note PI is used to imply phosphatidylinositol as well as propidium iodide in the text.

It is interesting to note that Jawsamycin is particularly potent against those Mold and Mucorales species that are resistant to available antifungals, voriconazole or echinocandin. However, it was not very effective against *C. neoformans* which naturally carries the equivalent of a T144A mutation in its predicted active site. The authors were also surprised at the lack of sensitivity of *C. albicans* to Jawsamycin since its Spt14/Gpi3 homolog shows an overall similarity of 57% to ScSpt14.

- The authors appear to be presuming that the Spt14/Gpi3 enzyme works in isolation here. A closer look at literature would reveal that even when ScSpt14 is considered to be the catalytic subunit, most other subunits of the phosphatidylinositol N-acetylglucosaminyl transferase are essential for the function of the enzyme complex. In this context the authors should (i) examine available literature to see whether the *C. albicans* enzyme or any of its subunits (especially those related to this study) have been shown to have any differences with those from *S. cerevisiae* and (ii) discuss whether differences in the subunit organization within the enzyme complex could alter active site conformation/ accessibility.

Jawsamycin was not toxic up to 50 M to human cell lines, HCT116, HEK293, HEPG2 and K562

(data for HCT116 alone are shown). Since PIG-A is not essential in mammalian cells and cell death due to Jawsamycin treatment could not have been monitored, GPI anchored proteins on the cell surface were monitored using FACS detection of FLAER labelled cells. As a positive control, PIG-A was inactivated using the CRISPR-Cas9 system in HCT116 cells and these cells stained negative for FLAER.

- The legends for Figure 3a do not match with the figure itself. The legends state, 'Cytotoxicity assessment by dose-response testingcontrol compound Dequalinium chloride against human HCT116 cells'. The figure shows that the control used is benzalkonium chloride. In fact, it would also help the reader if an explanation of why this was chosen as the control is given in the Methods or Results section.
- Are the data for the positive control also plotted on a log-scale?

Next, chemically modified variants of Jawsamycin were synthesized. Rather than begin by de novo synthesis, the authors use hydrolysis of Jawsamycin to obtain the cyclopropanated carboxylic acid which they then chemically modify to provide variations in nucleoside head groups and in the fatty acid tail. This is an interesting idea, one that should significantly reduce the time/ labor/ costs involved in the synthesis and should provide better yields than the more traditional approaches. This section too has several errors.

- The MEC values for Jawsamycin and its derivatives are presented in units of mg/ml in Figure 4, while Table 1 provides this in g/ml. So the data are three orders of magnitude apart. This requires a convincing explanation. I would like to see, either in the figure or figure legends or elsewhere in the manuscript, the actual MEC values for at least those compounds that are seen to be comparable to Jawsamycin in their effectiveness to inhibit fungal growth.
- Line 253: Shouldn't it be 5'-amino-5'-deoxy-5,6-dihydrouridine?
- Line 254: It should be cyclopropanated carboxylic acid not cyclopranated carboxylic acid.
- Line 268: The enzyme is not an UDP transferase.

Using an in vivo mouse model of invasive pulmonary mucormycosis (generated by R. delemar), they then tested the efficacy of Jawsamycin as an antifungal agent. Jawsamycin at 100 mg/kg appeared to be as effective as posaconazole in improving survival outcomes and reducing fungal burden in lungs and brain. Even a dose of Jawsamycin that was half the concentration of the clinically used dose of posaconazole (60 mg/kg) appeared to be as good as the latter in improving survival rates.

- 'Figure 5d, e) Tissue fungal burden on day 7 of mice treated like in experiment the experiment on panel b (n = 12) as defined by quantitative PCR.' What does this mean?
- Line 293: It should be 'tended' not 'trended'.

Discussion: While the discussion is reasonably adequate there are just too many grammatical and spelling errors in the text. Only a few are mentioned below. I suggest that a careful proofreading of the entire text be done.

- Line 342-343: 'The mechanism of action on Rhizopus spores' but the figure legends say that the data for *A. fumigatus* is presented.
- Line 311: The yeast/ fungal nomenclature is not PIG-Q, PIG-C, PIG-H and PIG-A. Instead, GPI1, GPI2, GPI15 and GPI3 should be used. In fact, this correction should be made in the title also, since PIG-A is used for the mammalian enzyme.
- Nowhere do the authors explain how many subunits form the functional phosphatidylinositol N-acetylglucosaminyl transferase and which are essential.
- Line 311-12: 'an unbiased resistance profiling studies ...' is grammatically wrong.
- Line 331-333: 'Thus, it appears that inhibition at the level of Spt14/PIG-A not only inhibits an essential function put furthermore exerts a dominant negative function'.
- Line 333-334: 'The downstream effect of compromised GPI biosynthesis result in cell-wall defects.' Wrong sentence construction.
- Line 346-348: 'Strikingly, the phenotype associated with Jawsamycin treatment is so dramatic, and many of the swollen conidia had apparent leakage of cellular content and consequent cell

death'. Incorrect sentence construction.

- Lines 352-353: 'With only few chemotypes available to modulate ever fewer nodes it is apparent that this pathway is underexplored and merits more attention'. What is meant by 'ever fewer nodes' in this context?
- Lines 353-355: 'It is to be hoped ...for additional screening campsins to identify more chemical starting points'. Spelling error.

Materials and Methods: The methodology is adequately described in most places. The controls used could be better explained for the comprehension of the reader who is not so familiar with these assays. Again, several minor language errors are noted. Some are identified below:

- Line 393: 'After Incubation...' should be 'After incubation...'
- Line 437: 'previously' should be 'previously'.
- Lines 460, 469: It should be 50,000 or 10,000 not 50'000 or 10'000.
- Line 470: 'For Jawsamycin effect testing, untreated cells were used to gate the Alexa Fluor FLAER positive cells...'

General comments:

- At many places the spacing between numbers and units is missing in the text and figure legends.
- The units for time used are interchangeably either hrs or h. Only 'h' should be used throughout.
- Some abbreviations are used without specifying their full forms the very first time that they are used.
- Nomenclature issues (protein/ gene/ species) must be carefully addressed.
- In general, names of chemicals are not proper nouns.

Graphical abstract: Overall, the graphical abstract correctly represents the salient point of the study. Again, some errors:

- The figure shows UDP-NAc. It should be UDP-GlcNAc. Also note, the manuscript text writes 'N-acetyl' as 'Nac' and not as 'NAC'.
- The PI shown appears to have a single lipid chain while it should have two.

Reviewer #2 (Remarks to the Author):

Major Claims/Findings: Morbidity and mortality from invasive fungal infections is increasing at a concerning rate driving the urgent need for new, more effective therapeutic strategies. To address the problem, this manuscript by Y. Fu et al. describes the high-throughput reporter-based screening of a 12,472-member natural product library to identify compounds capable of disrupting fungal glycosylphosphatidyl inositol (GPI) anchor biosynthesis. In addition to known inhibitors of this essential pathway in fungi, the authors identified the compound jawsamycin (FR-900848) as a reproducible, moderately potent hit. Subsequent chemogenomic profiling and an unbiased mutagenesis screen for resistant mutants provided compelling evidence for the catalytic subunit of the GlcNAc-Phosphatidylinositol complex (SPT14/PIG-A) as the target of jawsamycin responsible for its fungicidal activity. The compound did not inhibit GPI anchor biosynthesis in human cells as monitored by well-controlled FLAER staining experiments of multiple cancer lines. Consistent with this finding it was not cytotoxic to a panel of human cell lines at concentrations well over those required for antifungal activity against a broad panel of human pathogenic fungi, most notably multiple molds against which all current therapeutics have little or no efficacy. Routes to semi-synthesis of analogs for this complex natural product were developed, but unfortunately no improvements in potency or spectrum were obtained. Finally, the parent compound was evaluated in an immunocompromised mouse model of pulmonary mucormycosis where it showed significant antifungal activity at the doses and schedules tested.

Novelty: The activity of jawsamycin against filamentous fungi was reported decades ago, but until the work reported here, the mechanism of action for this unusual polycyclopropane–nucleoside hybrid was completely unknown. Targeting GPI anchor biosynthesis is a relatively new antifungal strategy, but an inhibitor of Gwt1, an essential acyl transferase in this pathway is now in late stage clinical development where it is showing considerable promise. No inhibitors of the first step in this pathway, the UDP-GlcNAc Phosphatidylinositol transferase complex, however, have been previously reported. Prior work defining the biosynthetic gene cluster responsible for production of jawsamycin by *Streptoverticillium* species as well as total synthesis of the compound have been previously reported. The semi-synthesis of novel analogs incorporating changes to either the long polycyclopropane tail or the nucleoside head group, however, is new.

Significance/Interest: The work will be of interest to both basic and translational scientists because the discovery of effective new antifungals, especially agents with activity against molds is being actively pursued, but has proven very difficult to achieve.

Recommendation: Overall, this is a well-conceived and well-executed study. The progression from screen hit through target identification and SAR to testing in culture and mouse model follows a well-established path. The data presented appear to be of reproducible, high quality with appropriate analysis of their statistical significance. Prior to publication, however, the following relatively minor concerns should be addressed:

1) The authors' claim that jawsamycin provides "an exciting starting point for structural modification to reach improvements of its derivatives for the treatment of devastating invasive fungal infections" seems over-stated. Instead, the very unusual, non-drug like structure of the compound and its inability to tolerate modification to either the head group or the long polycyclopropane tail could equally argue that the compound has served as a useful tool to identify a promising target for intervention, but is not a good lead for further development efforts. A more balanced appraisal of the compound's potential for development in the abstract and the discussion should be provided.

2) The cartoon representations of live and dead mice in the graphical abstract should be removed. They obscure the photomicrographs and give the impression that the images were somehow derived from samples taken from infected mice rather than in vitro cultures. A simple progression from target schematic to morphological effects in culture to Kaplan-Meier plot of survival in mice is suggested.

3) Line 91: "targets" should be "target"

4) Line 98: please define what "steep structure activity profile" means

5) Line 127: grammar of the sentence is incorrect and "fluorescence" should be replaced by "luminescence"

6) Line 133: IC30 should be replaced by IC50 to match with the data presented in Figure 1g

7) Line 209: How can MEC values be determined in the non-filamentous fungus *C. neoformans*? MIC would be more appropriate determination

8) Line 244 and following: The whole cell SAR described in this section does not distinguish between potential effects of modification on target affinity versus effects on intracellular accumulation (permeability and efflux). A discussion of this limitation in attempting to arrive at more active analogs should be provided.

9) Line 366: the phrase "add as you see fit!" seems to have been left in the text inadvertently

10) Line 571: Define "HPC"

11) The extreme sensitivity of Mucorales is intriguing. Do the authors think it arises as a result of increased compound accumulation in these organisms, increased target affinity or an extreme dependence on GPI anchor biosynthesis? Are these organisms also extremely sensitive to Gwt1 inhibitors? Some discussion of the issue seems warranted as it has important implications for further development of the compound.

12) Fig 1b: Why was no luminescence detected in the pellet fraction for the voriconazole-treated sample?

13) Fig 4: Are the analogs presented actually soluble at 100 mg/mL in RPMI culture medium to support the MEC values indicated? Seems unlikely given the need to use a nanoparticle suspension to achieve an adequate concentration for the dosing of mice. Should the values be microgram/mL? If so, why the discordance for *C. albicans* between Table 1 (MEC 1.2 microgram/mL) and this Figure (MEC < 0.5 microgram/mL)

14) Figure 5: Panel a: Abbreviations are not defined in the legend. To what does CsA refer? This abbreviation usually refers to cyclosporin A which the methods do not indicate was used in this experiment. To what does Cfz refer? This abbreviation often refers to ceftazidime, but again there is no mention of its use in the methods. Panels b and c: Statistical significance of differences in overall survival should be indicated in the figure

Reviewer #3 (Remarks to the Author):

Summary

Fu et al. present a study that identifies the natural product jawsamycin as a GPI synthesis inhibitor with antifungal activity in vitro and in vivo. GPI anchoring has recently been targeted for antifungal activity and the Gwt1 inhibitor APX001 exhibits broad spectrum antifungal activity and has progressed to clinical trials. Thus, GPI linkage is an excellent and emerging antifungal target. The authors employed an elegant screen to select compounds that specifically promoted the release of a GPI-anchored probe in *S. cerevisiae*. In addition to known GPI inhibitors such as APX001 (E1210), the natural product jawsamycin increased GPI-linked probe released in the media. The broad-spectrum antifungal activity of jawsamycin was first reported in 1990 but the target and mechanism of action remained unknown. Here, through the HIP-HOP system, Fu et al. identify PIG-A as the target of jawsamycin. Additionally, they generate resistant mutants with point mutations in PIG-A to confirm the target. One of the major pitfalls of novel antifungal drug development is human toxicity. Jawsamycin demonstrates high fungal selectivity and little toxicity in multiple mammalian cell lines. To better understand the structure-activity relationship with 3 chemically accessible moieties of jawsamycin, a small library of 20 jawsamycin analogs was generated and screened for antifungal activity. Although all of the analogs exhibited a marked decrease in antifungal activity, the authors suggest that they have developed novel protocols for the large-scale generation of analogs and will be able to screen many more analogs to complement this smaller SAR study. Finally, in a murine model of mucormycosis, jawsamycin modestly increased animal survival in two different dosing strategies compared to vehicle treated animals.

The need for novel antifungals is very clear and the authors clearly present this unmet medical need in their introduction. Because there are only 4 classes of antifungal drugs used clinically to treat systemic fungal infections, compounds that utilize novel targets are even more valuable. GPI

targeting has proven to be an effective strategy in the case of APX001 and additional compounds within that class, though no longer novel, are still valuable. The authors additionally developed a platform for the large-scale synthesis and modification of jawsamycin that could lead to a better understanding of the molecular MOA of jawsamycin in the future.

Major Issues

- The authors demonstrate that jawsamycin exhibits excellent, broad spectrum antifungal activity with low MICs. However, the synergy with current antifungal drugs was not tested. This would be especially interesting in the context of the Mucorales given their robust resistance to many antifungal agents to explore potentiation of an otherwise ineffective antifungal. The authors also showed through HOP data that jawsamycin is synthetic lethal with ER stress. This could also be demonstrated chemically.
- The murine infection model showed a modest increase in survival. Could the dosing be adjusted (higher, more frequent) to increase the in vivo efficacy? If the toxicity limits were not reached at 100 mg/kg jawsamycin are there issues with solubility that would prevent higher doses? Have other infection models been tested?

Minor Issues

- Additional references could be placed in the intro for the current GPI inhibitors
- Grammar, punctuation errors throughout
- Could describe the sequence divergence between human and fungi in more detail to emphasize opportunity for selectivity in the introduction
- Emphasize that jawsamycin was found to exhibit broad spectrum antifungal activity previously
- Line 133 says "IC30 of ~800 nM" should be changed to IC50

To the editor and the reviewers,

We would like to thank all three reviewers and the editor for the careful evaluation of our manuscript. It is obvious by the detailed feedback that all reviewers have devoted significant amount of time to read the manuscript and propose suggestions and corrections. It is a privilege to have scientific colleagues that go the extra mile to help others to disseminate their research in the best possible fashion. Thank you!

As you will see, we have considered your feedback carefully and incorporated all your corrections.

Due to the current COVID-19 situation and the suspended lab activities we have not been able to perform additional experiments (e.g. drug combination testing). As >90% of the comments concerned text edits we have decided to resubmit. We count on the understanding of the reviewers in these difficult times.

The authors

Reviewer #1:

Systemic fungal infections are responsible for high rates of mortality and are a serious public health challenge. Since they are eukaryotes like the host, it has been difficult to find drugs that specifically target the pathogen. Hence, a very limited repertoire of drugs is available to treat fungal infections. Also, as use of these antifungals have become widespread, the pathogens responsible for disease too have begun to shift to more drug resistant fungal species on which the current drugs have poor efficacy. Thus, the search for new, effective antifungals is urgent. The present manuscript studies the possibility of using Jawsamycin (a natural product identified during the course of this study as a fungal-specific GPI biosynthetic inhibitor) as a new antifungal with potential clinical applications.

The study is timely, well designed/executed and likely to be of interest to a wide audience. I would recommend publication but after major revisions.

The authors provide a reasonably good Introduction to the study and to the challenges of finding new fungal inhibitors/ antifungals. The results are potentially very exciting. However, there are several errors. The most serious of these is the fact that the figure legends either do not match with the text or the relevant figure or both.

The authors begin by performing a high-throughput screen of 12472 compounds from the Novartis library of natural products to test their effectiveness against the GPI anchor biosynthetic pathway in fungi. Their rationale for choosing this pathway as a drug target is quite convincing: targeting the GPI anchor biosynthesis should also subject the pathogen to cellular stress, compromise its cell wall integrity and expose the cell wall beta- glucans to the host's immune response pathway. There are also some differences between the mammalian and fungal GPI biosynthetic enzymes, which they hoped to be able to exploit to make the drug specific for the pathogen.

A quick method for screening the compounds was developed. For this purpose, the *G. princeps* luciferase gene was fused to the GPI anchoring signal sequence of *Candida albicans* PGA59 and attached to the inducible GAL1/10 promoter and expressed in *S. cerevisiae*. Known antifungals, voriconazole and amphotericin B (which do not target GPI biosynthesis), as well as the GPI inhibitor, E1210 (which inhibits the Gwt1 enzyme), were used to show that the assay methodology works and is specific for inhibition of GPI biosynthesis. Inhibition of GPI biosynthesis alone results in the luciferase enzyme being released into the medium rather than remaining bound to the cell surface. This was then used to screen the natural products library, which led to the identification of an oligocyclopropyl-containing natural product, named Jawsamycin. This was further validated by using a fresh batch of Jawsamycin purified from the producer strain, *Streptomyces luteoverticillatus*, and yielded an IC50 of 7 M in the assay. Also, since Jawsamycin inhibited growth of *S. cerevisiae*, it was next used in HIP/HOP assays with *S. cerevisiae* to identify potential targets.

- The rationale for using 10 μ M for voriconazole, and 1 μ M for amphotericin B and E-1210 in the experiment needs to be explained (Figure 1b).

These values roughly correspond to IC₅₀ concentrations (rounded for ease of pipetting) determined in pre-experiments. This information has been added into the figure legend.

- In the text, IC30 for growth inhibition by Jawsamycin is stated to be 800 nM. No details are given in the figure legends, but in Figure 1g, IC50 is mentioned instead of IC30. Which should it be?

Thank you for spotting this inconsistency. We have corrected this to IC₅₀.

Also, details of the experiment and how growth was monitored need to be given either here or in the Methods section.

This has been inserted.

- What do the error bars in Figure 1 represent?

The depict the standard error and the dots the mean of two replicates. This information has been inserted into the figure legend.

- Nomenclature is a problem at several places in the text. For example, is it *Streptomyces luteovorticillatus* or *Streptovorticillium fervens*?

*Thank you for spotting this inconsistency. Jawsamycin was isolated from *Streptomyces luteovorticillatus* and this has been corrected in the text.*

Two independent HIP experiments identified genes of the enzyme complex involved in the first step of GPI biosynthesis, GPI1, GPI2, GPI3 (SPT14) and GPI15, as possible targets with good z-scores. The growth inhibition of individual strains with Jawsamycin was retested for confirmation. Not surprisingly, genes of the unfolded protein response pathway (HAC1 and IRE1) and the cell wall integrity pathway (MID2/JIP3, RLM1) were synthetic lethal with GPI biosynthesis inhibition by Jawsamycin and showed up in HOP experiments.

- The experiments also identified the pre-mRNA splicing component, MUD2, as synthetically lethal with the genes of the GPI biosynthetic pathway (as seen in the figure). A discussion of its relevance is required. *A short section has been added to the discussion. Interestingly, MUD2 only scores chemical/synthetic lethality with inhibition of SPT14/PIG3 but not GWT1 or MCD4. We account this to the fact that only the SPT14/PIG3 gene contains an intron and hence is dependent on the yeast splicing machinery.*

- The data given in Supplementary Figure S1 does not mention the value of 'n'. Also, it is not clear whether IC50 or IC30 was tested. OD600 should be OD600 nm.

The value of n has been added to the figure legend and the y-axis description updated to "OD_{600 nm}". As an entire concentration range was tested (concentrations indicated on the x-axis) we refrain to mention anything regarding the IC₅₀/IC₃₀.

- The gene name, SPT14, should be replaced by GPI3 to keep the nomenclature uniform for the GPI genes. *Thank you for bringing this up. We have considered this. However, the standard *S. cerevisiae* gene name proposed by the Saccharomyces Genome Consortium is SPT14. We have decided to introduce the GPI3 alias name in the title and the abstract and introduction but then stick to the standard name.*

As our tests did not reveal activity against the human PIG-A homolog naming it SPT14/PIG-A may be misleading and we have removed this.

In addition, (as also requested by the second reviewer) we have corrected nomenclature for genes and gene products to the standard nomenclature for fungi (genes: italicized, with all letters in uppercase; protein: first letter in uppercase, nonitalic) and mammalian cells (genes: italicized, with all letters in uppercase; protein: all uppercase, nonitalic).

- It is rather surprising that GPI19, which is an essential gene, did not get identified as a target. At least a mention of the fact that the enzyme complex has two additional subunits which were not identified in the HIP assay should be mentioned.

GPI19 does show moderate hypersensitivity in both repeats and ERI1 was not resolved as this deletion strain is not part of the original yeast deletion collection. This information has been added into the results section and the GPI19 stain labeled in Figure 2a.

Using an unbiased mutagenesis screen, mutations were identified in SPT14 (heterozygous H106Y mutation, V76F mutation and T144A mutation) that provided resistance to Jawsamycin. No mutations were identified in GPI1, GPI2 or GPI15. To confirm that the mutations led to resistance, these were then introduced in the wild type SPT14 gene. A 10-100 fold shift in the IC50 confirmed that the mutations conferred resistance to Jawsamycin. The mutations had no effect on sensitivity to clotrimazole, an Erg11 inhibitor. Strong propidium iodide staining suggests that the action is cidal in nature.

- Figure 2c: The grey domain in the figure is mentioned as 'predicted PI-n-acetylglucosaminyltransferase subunit'. Firstly, this is not a subunit. Perhaps the authors meant to identify it as the predicted catalytic domain. Also, it should be PI-N-acetylglucosaminyltransferase rather than 'PI-n-acetylglucosaminyltransferase subunit'.

Thank you for spotting this. We have corrected this as suggested.

- Legends for Figure 2d state: 'd) Frequency, IC50 concentrations and fold resistance of identified mutant clones as calculated by using the curves depicted in panel E and F.' No panels E and F exist in the figure. It should be panel 'e' since only the data for Jawsamycin have been listed. The data from panel 'f' too should be given or the range summarized in the legends.

We have changed "E" to lower case "e" and inserted a short statement for the clotrimazole curves into the figure legend. As requested for Figure S1, we have also inserted the information on the value of 'n'.

- Was PI staining of Jawsamycin-treated cells done for *A. fumigatus* or *R. oryzae*? The figure legends (Figure 2g) say that the data presented is for *A. fumigatus*, while the text mentions *R. oryzae*.

*The PI staining was done with *R. oryzae*. We have corrected the figure legends.*

- PI staining of at least one species from Mold and one from Mucorales would have been useful, given that the cell wall architectures are not the same in these cases.

The purpose of PI staining in this study is mainly to show the fungicidal activity of Jawsamycin and accessibility of nucleic acid in the dead cells upon the compound treatment. We do notice that cell wall has relatively weak PI staining, but this type of staining is not the objective of this study.

- Note PI is used to imply phosphatidylinositol as well as propidium iodide in the text.

Thank you for pointing this out. We no longer used PI as an abbreviation in the text (except in the context of GPI) and use "propidium iodine" in Figure 2g.

It is interesting to note that Jawsamycin is particularly potent against those Mold and Mucorales species that are resistant to available antifungals, voriconazole or echinocandin. However, it was not very effective against *C. neoformans* which naturally carries the equivalent of a T144A mutation in its predicted active site. The authors were also surprised at the lack of sensitivity of *C. albicans* to Jawsamycin since its Spt14/Gpi3 homolog shows an overall similarity of 57% to ScSpt14.

- The authors appear to be presuming that the Spt14/Gpi3 enzyme works in isolation here. A closer look at literature would reveal that even when ScSpt14 is considered to be the catalytic subunit, most other subunits of the phosphatidylinositol N-acetylglucosaminyl transferase are essential for the function of the enzyme complex. In this context the authors should (i) examine available literature to see whether the *C. albicans* enzyme or any of its subunits (especially those related to this study) have been shown to have any differences with those from *S. cerevisiae* and (ii) discuss whether differences in the subunit organization within the enzyme complex could alter active site conformation/ accessibility.

*The reviewer has brought up a very good point that Spt14/Gpi3 works in a large enzymatic complex. It's intriguing that even though the composition of the phosphatidylinositol N-acetylglucosaminyl transferase (GPI-GnT) in *S. cerevisiae* and *C. albicans* are identical, and both are composed of Gpi1, Gpi2, Gpi3, Gpi15, Gpi19, and Eri1, there are subtle differences in interactions between the subunits, and with other proteins such as Ras. In this study, we used *S. cerevisiae* as a tool organism for compound screening as well as for generating the target hypothesis. We are not suggesting that any difference in the enzyme complex contributes to any species difference in susceptibility to Jawsamycin.*

Jawsamycin was not toxic up to 50 M to human cell lines, HCT116, HEK293, HEPG2 and K562 (data for HCT116 alone are shown). Since PI-G-A is not essential in mammalian cells and cell death due to Jawsamycin treatment could not have been monitored, GPI anchored proteins on the cell surface were

monitored using FACS detection of FLAER labelled cells. As a positive control, PIG-A was inactivated using the CRISPR-Cas9 system in HCT116 cells and these cells stained negative for FLAER.

- The legends for Figure 3a do not match with the figure itself. The legends state, 'Cytotoxicity assessment by dose-response testingcontrol compound Dequalinium chloride against human HCT116 cells'. The figure shows that the control used is benzalkonium chloride. In fact, it would also help the reader if an explanation of why this was chosen as the control is given in the Methods or Results section.

Thank you for spotting this inconsistency. We have used benzalkonium chloride and corrected the figure legend. We have also inserted a sentence into the Materials & Methods section why this was chosen as cytotoxic, positive control (it is inexpensive, stable, results in a steep inhibition curve, does not interfere with luminescence readouts and is less hazardous for the operator than for example cycloheximide).

- Are the data for the positive control also plotted on a log-scale?

Yes, although depicted in % it is still log scale. Unfortunately, benzalkonium chloride (due to different alkyl chain lengths) is sold as a polymeric mixture. This prohibits to display its absolute concentration and this is why we use %.

Next, chemically modified variants of Jawsamycin were synthesized. Rather than begin by de novo synthesis, the authors use hydrolysis of Jawsamycin to obtain the cyclopropanated carboxylic acid which they then chemically modify to provide variations in nucleoside head groups and in the fatty acid tail. This is an interesting idea, one that should significantly reduce the time/ labor/ costs involved in the synthesis and should provide better yields than the more traditional approaches. This section too has several errors.

- The MEC values for Jawsamycin and its derivatives are presented in units of mg/ml in Figure 4, while Table 1 provides this in g/ml. So the data are three orders of magnitude apart. This requires a convincing explanation. I would like to see, either in the figure or figure legends or elsewhere in the manuscript, the actual MEC values for at least those compounds that are seen to be comparable to Jawsamycin in their effectiveness to inhibit fungal growth.

Thanks for spotting this error. There was a formatting issue in Figure 4, the unit should be microgram / mL. We have corrected this.

- Line 253: Shouldn't it be 5'-amino-5'-deoxy-5,6-dihydrouridine?

Yes, and it reads like this in the .doc document. We do not understand why this got corrupted when converted to .pdf. We have typed it once more and hope it will correctly convert this time.

- Line 254: It should be cyclopropanated carboxylic acid not cyclopranated carboxylic acid.

Yes. We have corrected this.

- Line 268: The enzyme is not an UDP transferase.

We have corrected this to "UDP-glycosyltransferase"

Using an in vivo mouse model of invasive pulmonary mucormycosis (generated by R. delemar), they then tested the efficacy of Jawsamycin as an antifungal agent. Jawsamycin at 100 mg/kg appeared to be as effective as posaconazole in improving survival outcomes and reducing fungal burden in lungs and brain. Even a dose of Jawsamycin that was half the concentration of the clinically used dose of posaconazole (60 mg/kg) appeared to be as good as the latter in improving survival rates.

- 'Figure 5d, e) Tissue fungal burden on day 7 of mice treated like in experiment the experiment on panel b (n = 12) as defined by quantitative PCR.' What does this mean?

We apologize for the ambiguity in this legend. In summary, the tissue fungal burden were processed on day 7 post infection from 12 mice/group. Determination of the tissue burden was carried out by real-time qPCR as described in the Method section. This is now described in the figure legend.

- Line 293: It should be 'tended' not 'trended'.

Yes, we have corrected this.

Discussion: While the discussion is reasonably adequate there are just too many grammatical and spelling errors in the text. Only a few are mentioned below. I suggest that a careful proofreading of the entire text be done.

Thank you. We have asked an English native speaker to proofread the discussion and have incorporated the suggested edits.

- Line 342-343: 'The mechanism of action on Rhizopus spores' but the figure legends say that the data for *A. fumigatus* is presented.

*Thanks for spotting this error in the figure legend. It should be *R. oryzae* and it has been corrected.*

- Line 311: The yeast/ fungal nomenclature is not PIG-Q, PIG-C, PIG-H and PIG-A. Instead, GPI1, GPI2, GPI15 and GPI3 should be used. In fact, this correction should be made in the title also, since PIG-A is used for the mammalian enzyme.

*We have replaced the mammalian gene names as suggested. As outlined to reviewer 1 we propose to use *SPT14* as it is the standard name proposed by the *Saccharomyces Genome Database* consortium. But we give reference to its *GPI3* alias in the title, abstract and introduction.*

- Nowhere do the authors explain how many subunits form the functional phosphatidylinositol N-acetylglucosaminyl transferase and which are essential.

This has now been inserted in the results section in the chemogenomic profiling paragraph when we describe the HIP HOP findings.

- Line 311-12: 'an unbiased resistance profiling studies ...' is grammatically wrong.

We have changed to "unbiased resistance profiling studies".

- Line 331-333: 'Thus, it appears that inhibition at the level of Spt14/PIG-A not only inhibits an essential function put furthermore exerts a dominant negative function'.

We have corrected "put" to "but".

- Line 333-334: 'The downstream effect of compromised GPI biosynthesis result in cell-wall defects.'

Wrong sentence construction.

Corrected.

- Line 346-348: 'Strikingly, the phenotype associated with Jawsamycin treatment is so dramatic, and many of the swollen conidia had apparent leakage of cellular content and consequent cell death'.

Incorrect sentence construction.

We have deleted this phrase as it is repetitive with previous content in this section.

- Lines 352-353: 'With only few chemotypes available to modulate ever fewer nodes it is apparent that this pathway is underexplored and merits more attention'. What is meant by 'ever fewer nodes' in this context?

*We have changed to "...just two targets (*GWT1* and *MCD4*) this node...".*

- Lines 353-355: 'It is to be hoped ...for additional screening campaigns to identify more chemical starting points'. Spelling error.

We have deleted the incorrect "for".

Materials and Methods: The methodology is adequately described in most places. The controls used could be better explained for the comprehension of the reader who is not so familiar with these assays. Again, several minor language errors are noted. Some are identified below:

- Line 393: 'After Incubation...' should be 'After incubation...'.
Corrected.

Corrected.

- Line 437: 'previously' should be 'previously'.

Corrected.

- Lines 460, 469: It should be 50,000 or 10,000 not 50'000 or 10'000.

Corrected.

- Line 470: 'For Jawsamycin effect testing, untreated cells were used to gate the Alexa Fluor FLAER positive cells...'.
We have rephrased this section.

General comments:

- At many places the spacing between numbers and units is missing in the text and figure legends.
We have proofread the manuscript and hope to have spotted and corrected all these errors.

- The units for time used are interchangeably either hrs or h. Only 'h' should be used throughout.
Corrected.

- Some abbreviations are used without specifying their full forms the very first time that they are used.
We have corrected this when proofreading the manuscript. Each abbreviation now is introduced when used for the first time.

- Nomenclature issues (protein/ gene/ species) must be carefully addressed.
We have corrected nomenclature for genes and gene products to the standard nomenclature for fungi (genes: italicized, with all letters in uppercase; protein: first letter in uppercase, nonitalic) and mammalian cells (genes: italicized, with all letters in uppercase; protein: all uppercase, nonitalic). Species are written in italic letters with a first capital letter for the Genus.

- In general, names of chemicals are not proper nouns.
We have corrected this throughout the manuscript and the figures (e.g. jawsamycin, posaconazol, benzalkonium chloride, etc.).

Graphical abstract: Overall, the graphical abstract correctly represents the salient point of the study. Again, some errors:

- The figure shows UDP-NAc. It should be UDP-GlcNAc. Also note, the manuscript text writes 'N-acetyl' as 'Nac' and not as 'NAC'.

This has been corrected in the graphical abstract and throughout the text.

- The PI shown appears to have a single lipid chain while it should have two.
It had two chains but due to resolution they overlapped. We have spaced them further apart.

Reviewer #3 (Remarks to the Author):

Major Claims/Findings: Morbidity and mortality from invasive fungal infections is increasing at a concerning rate driving the urgent need for new, more effective therapeutic strategies. To address the problem, this manuscript by Y. Fu et al. describes the high-throughput reporter-based screening of a 12,472-member natural product library to identify compounds capable of disrupting fungal glycosylphosphatidyl inositol (GPI) anchor biosynthesis. In addition to known inhibitors of this essential pathway in fungi, the authors identified the compound jawsamycin (FR-900848) as a reproducible, moderately potent hit. Subsequent chemogenomic profiling and an unbiased mutagenesis screen for resistant mutants provided compelling evidence for the catalytic subunit of the GlcNAc-Phosphatidylinositol complex (SPT14/PIG-A) as the target of jawsamycin responsible for its fungicidal activity. The compound did not inhibit GPI anchor biosynthesis in human cells as monitored by well-controlled FLAER staining experiments of multiple cancer lines. Consistent with this finding it was not cytotoxic to a panel of human cell lines at concentrations well over those required for antifungal activity against a broad panel of human pathogenic fungi, most notably multiple molds against which all current therapeutics have little or no efficacy. Routes to semi-synthesis of analogs for this complex natural product were developed, but unfortunately no improvements in potency or spectrum were obtained. Finally, the parent compound was evaluated in an immunocompromised mouse model of pulmonary mucormycosis where it showed significant antifungal activity at the doses and schedules tested.

Novelty: The activity of jawsamycin against filamentous fungi was reported decades ago, but until the work reported here, the mechanism of action for this unusual polycyclopropane-nucleoside hybrid was

completely unknown. Targeting GPI anchor biosynthesis is a relatively new antifungal strategy, but an inhibitor of Gwt1, an essential acyl transferase in this pathway is now in late stage clinical development where it is showing considerable promise. No inhibitors of the first step in this pathway, the UDP-GlcNAc Phosphatidylinositol transferase complex, however, have been previously reported. Prior work defining the biosynthetic gene cluster responsible for production of jawsamycin by *Streptoverticillium* species as well as total synthesis of the compound have been previously reported. The semi-synthesis of novel analogs incorporating changes to either the long polycyclopropane tail or the nucleoside head group, however, is new.

Significance/Interest: The work will be of interest to both basic and translational scientists because the discovery of effective new antifungals, especially agents with activity against molds is being actively pursued, but has proven very difficult to achieve.

Recommendation: Overall, this is a well-conceived and well-executed study. The progression from screen hit through target identification and SAR to testing in culture and mouse model follows a well-established path. The data presented appear to be of reproducible, high quality with appropriate analysis of their statistical significance. Prior to publication, however, the following relatively minor concerns should be addressed:

1) The authors' claim that jawsamycin provides "an exciting starting point for structural modification to reach improvements of its derivatives for the treatment of devastating invasive fungal infections" seems over-stated. Instead, the very unusual, non-drug like structure of the compound and its inability to tolerate modification to either the head group or the long polycyclopropane tail could equally argue that the compound has served as a useful tool to identify a promising target for intervention, but is not a good lead for further development efforts. A more balanced appraisal of the compound's potential for development in the abstract and the discussion should be provided.

We agree with the reviewer's view. We have replaced these sentences by more moderate language both in the abstract as well as in the discussion and have deleted the term "exciting".

2) The cartoon representations of live and dead mice in the graphical abstract should be removed. They obscure the photomicrographs and give the impression that the images were somehow derived from samples taken from infected mice rather than in vitro cultures. A simple progression from target schematic to morphological effects in culture to Kaplan-Meier plot of survival in mice is suggested.

We have removed the overlaps and follow the reviewers advise to go from (left) target/mechanism to (middle) in vitro to (right) in vivo. We propose to keep the schematic representation of the mice to make it obvious that this was the in vivo species. Only showing the survival plot may suggest that this is clinical data from human patients.

3) Line 91: "targets" should be "target"

This has been corrected.

4) Line 98: please define what "steep structure activity profile" means

We have rephrased as follows: "...identifying a thigh structure activity relationship".

5) Line 127: grammar of the sentence is incorrect and "fluorescence" should be replaced by "luminescence"

Many thanks for the attention to detail! We have replaced fluorescence and rephrased the entire sentence.

6) Line 133: IC30 should be replaced by IC50 to match with the data presented in Figure 1g.

This has been corrected.

7) Line 209: How can MEC values be determined in the non-filamentous fungus *C. neoformans*? MIC would be more appropriate determination

The MEC is defined as the concentration that resulted in a prominent reduction of growth, and most often also referred as MIC-2.

8) Line 244 and following: The whole cell SAR described in this section does not distinguish between potential effects of modification on target affinity versus effects on intracellular accumulation (permeability and efflux). A discussion of this limitation in attempting to arrive at more active analogs should be provided.

We agree with the reviewer's view and have inserted a sentence in the discussion.

9) Line 366: the phrase "add as you see fit!" seems to have been left in the text inadvertently
Indeed. We have removed this.

10) Line 571: Define "HPC".
HPC refers to hydroxypropyl cellulose. We now have defined the abbreviation the first time this term is used.

11) The extreme sensitivity of Mucorales is intriguing. Do the authors think it arises as a result of increased compound accumulation in these organisms, increased target affinity or an extreme dependence on GPI anchor biosynthesis? Are these organisms also extremely sensitive to Gwt1 inhibitors? Some discussion of the issue seems warranted as it has important implications for further development of the compound.

The reviewer brought up a very interesting point. From our studies, we did not observe similar sensitivity of Mucorales to Gwt1 inhibitors. Given jawsamycin is a natural product, it can be speculated that the producing organism might use this strong potency to gain competitive advantage against the surrounding micro-organisms. Although we don't have the exact information of the origin of the jawsamycin producing strains, we think it's likely that they were isolated from soil samples. This might correlate well with the fact that most Mucorales species are associated with soil and plants. Interestingly, even though most Aspergillus species are not highly susceptible to jawsamycin, A. flavus, which is most frequently associated with plants, is highly sensitive to this nature product. But as this is pure speculation we prefer not to elaborate on this in the discussion.

12) Fig 1b: Why was no luminescence detected in the pellet fraction for the voriconazole-treated sample?
Thank you very much for pointing this out! For some reason the bar for voriconazole did get lost when exporting the figure from Tibco Spotfire as .emf file. We have repeated and placed a new figure that now contains all the data.

13) Fig 4: Are the analogs presented actually soluble at 100 mg/mL in RPMI culture medium to support the MEC values indicated? Seems unlikely given the need to use a nanoparticle suspension to achieve an adequate concentration for the dosing of mice. Should the values be microgram/mL? If so, why the discordance for C. albicans between Table 1 (MEC 1.2 microgram/mL) and this Figure (MEC < 0.5 microgram/mL)

Thanks for spotting this formatting error. The unit should be microgram / mL. We have corrected this.

14) Figure 5: Panel a: Abbreviations are not defined in the legend. To what does CsA refer? This abbreviation usually refers to cyclosporin A which the methods do not indicate was used in this experiment. To what does Cfz refer? This abbreviation often refers to ceftazidime, but again there is no mention of its use in the methods. Panels b and c: Statistical significance of differences in overall survival should be indicated in the figure

We apologize for the ambiguity in this Figure. The reviewer is correct that CsA usually refers to cyclosporin A which was not used in this study. As stated in the methods, mice were immunosuppressed with cyclophosphamide (CP) and cortisone acetate (CA). Also, the reviewer is correct that Cfz stands for ceftazidime which was used to prevent bacterial superinfection. All these abbreviations are now explained in the figure legend and treating of mice with antibiotics to prevent bacterial infection is added to the Methods section. Finally, we added the statistical significance in panels b and c.

Reviewer #3 (Remarks to the Author):

Summary

Fu et al. present a study that identifies the natural product jawsamycin as a GPI synthesis inhibitor with antifungal activity in vitro and in vivo. GPI anchoring has recently been targeted for antifungal activity and the Gwt1 inhibitor APX001 exhibits broad spectrum antifungal activity and has progressed to clinical trials. Thus, GPI linkage is an excellent and emerging antifungal target. The authors employed an elegant screen to select compounds that specifically promoted the release of a GPI-anchored probe in S.

cerevisiae. In addition to known GPI inhibitors such as APX001 (E1210), the natural product jawsamycin increased GPI-linked probe released in the media. The broad-spectrum antifungal activity of jawsamycin was first reported in 1990 but the target and mechanism of action remained unknown. Here, through the HIP-HOP system, Fu et al. identify PIG-A as the target of jawsamycin. Additionally, they generate resistant mutants with point mutations in PIG-A to confirm the target. One of the major pitfalls of novel antifungal drug development is human toxicity. Jawsamycin demonstrates high fungal selectivity and little toxicity in multiple mammalian cell lines. To better understand the structure-activity relationship with 3 chemically accessible moieties of jawsamycin, a small library of 20 jawsamycin analogs was generated and screened for antifungal activity. Although all of the analogs exhibited a marked decrease in antifungal activity, the authors suggest that they have developed novel protocols for the large-scale generation of analogs and will be able to screen many more analogs to complement this smaller SAR study. Finally, in a murine model of mucormycosis, jawsamycin modestly increased animal survival in two different dosing strategies compared to vehicle treated animals.

The need for novel antifungals is very clear and the authors clearly present this unmet medical need in their introduction. Because there are only 4 classes of antifungal drugs used clinically to treat systemic fungal infections, compounds that utilize novel targets are even more valuable. GPI targeting has proven to be an effective strategy in the case of APX001 and additional compounds within that class, though no longer novel, are still valuable. The authors additionally developed a platform for the large-scale synthesis and modification of jawsamycin that could lead to a better understanding of the molecular MOA of jawsamycin in the future.

Major Issues

- The authors demonstrate that jawsamycin exhibits excellent, broad spectrum antifungal activity with low MICs. However, the synergy with current antifungal drugs was not tested. This would be especially interesting in the context of the Mucorales given their robust resistance to many antifungal agents to explore potentiation of an otherwise ineffective antifungal. The authors also showed through HOP data that jawsamycin is synthetic lethal with ER stress. This could also be demonstrated chemically.

The reviewer has brought up a great point of testing synergies of Jawsamycin with the current antifungal drugs. Due to the COVID-19 situation and suspended lab activities we have not been able to execute this experiment.

- The murine infection model showed a modest increase in survival. Could the dosing be adjusted (higher, more frequent) to increase the in vivo efficacy? If the toxicity limits were not reached at 100 mg/kg jawsamycin are there issues with solubility that would prevent higher doses? Have other infection models been tested?

Dosing at higher concentrations was not tested due to solubility limits of jawsamycin with the chosen formulation. No other infection models have been tested for this study.

Minor Issues

- Additional references could be placed in the intro for the current GPI inhibitors

We have added references to:

Tsukahara, K. et al. 2003 and McLellan, C.A. et al. 2012 for the discovery of the original GWT1 inhibitors, and Hong, Y. et al. 1999 and Mann, P.A. et al. 2015 for the MCD4 compounds

For the medicinal chemistry and in vivo testing of the E1210 and APX001 Zhao, M. et al. 2018 and Viriyakosol, S. et al. 2019.

We hope that in combination with the excellent review by Mutz and Roemer, 2016 this gives adequate credit to our scientific colleagues.

- Grammar, punctuation errors throughout

The manuscript has been proofread by an English native speaker and various corrections and edits implemented.

- Could describe the sequence divergence between human and fungi in more detail to emphasize opportunity for selectivity in the introduction

As pointed out, overall there is significant conservation of individual genes across species. As a single amino acid change can result in ~100 fold IC50 shift (Figure 2e) we prefer not to make any statements on selectivity. In previous work (doi: 10.1016/j.chom.2012.04.015), we were amazed by a ~1000 fold selectivity between human and pathogen by the natural product cladosporin despite an almost perfectly conserved binding pocket.

- Emphasize that jawsamycin was found to exhibit broad spectrum antifungal activity previously
This has been included and referenced.

- Line 133 says “IC30 of ~800 nM” should be changed to IC50
This has been corrected.